

# libcloudph++ 1.1: aqueous phase chemistry extension of the Lagrangian cloud microphysics scheme

Anna Jaruga and Hanna Pawlowska

Institute of Geophysics, Faculty of Physics, University of Warsaw, Warsaw, Poland

**Correspondence:** Anna Jaruga (ajaruga@igf.fuw.edu.pl)

**Abstract.** This paper introduces a new scheme available in the library of algorithms for representing cloud microphysics in numerical models named *libcloudph++*. The scheme extends the Lagrangian microphysics scheme available in *libcloudph++* to the aqueous phase chemical processes occurring within cloud droplets. The representation of chemical processes focuses on the aqueous phase oxidation of the dissolved $SO_2$ by $O_3$ and $H_2O_2$. The Lagrangian Microphysics and Chemistry (LMC)

scheme allows tracking the changes in the cloud condensation nuclei (CCN) distribution caused by both collisions between cloud droplets and aqueous phase oxidation.

The scheme is implemented in C++ and equipped with bindings to Python which allow reusing the created scheme from models implemented in other programming languages. The scheme can be used on either CPU or GPU, and is distributed under the GPL3 license. Here, the LMC scheme is tested in a simple 0-dimensional adiabatic parcel model and then used in a

2-dimensional prescribed flow framework. The results are discussed with the focus on changes to the CCN sizes and compared with other model simulations discussed in the literature.

## 1 Introduction

*libcloudph++* is an open-source library of schemes for representing cloud microphysics in numerical models. It was first

introduced in Arabas et al. (2015) where the authors present the different microphysics schemes available in the library, show its user interface, and discuss its performance. The flagship component of *libcloudph++* is the Lagrangian (i.e. particle tracking or particle based) microphysics scheme. The scheme resolves the evolution of aerosol, cloud droplet, and rain drop[1] size spectrum. It allows representing from the first principles cloud microphysical processes and is especially well suited to track changes in the CCN size distribution that are caused by clouds (i.e. cloud-aerosol processing). The scheme can be used in models of any

dimensionality or dynamical core, and can be run on both CPU and GPU. The open-source availability of the code, its clearly defined user interface, and the separation of concerns employed when designing *libcloudph++* core code enable its usage and further development.

---

[1] For convenience cloud droplets and rain drops will be often labeled together as water drops



This paper documents the extension of the Lagrangian microphysics scheme with a numerical scheme that represents aqueous phase chemical reactions inside cloud droplets. The representation of chemical reactions includes only the aqueous phase processes (i.e., no gas phase chemical reactions) and revolves around oxidation of sulfur dissolved in water drops to sulfate. Two reaction paths are considered – oxidation by ozone and by hydrogen peroxide. In total, six trace gases are included in the

chemistry description: sulfur dioxide ($SO_2$), ozone ($O_3$), hydrogen peroxide ($H_2O_2$), carbon dioxide ($CO_2$), nitric acid ($HNO_3$), and ammonia ($NH_3$). Their dissolution, and if applicable, dissociation is resolved.

Aqueous phase oxidation of sulfur is an important chemical reaction. Sulfur is emitted to the atmosphere by phytoplankton in the oceans as dimethyl sulfide and is then oxidized in the gas phase to sulfur dioxide trace gas ($SO_2$). $SO_2$ is also emitted

by anthropogenic and volcanic activity. The gas phase $SO_2$ is oxidized in a matter of days either by the gas phase or aqueous phase reactions, see the review by Faloona (2009). The aqueous phase oxidation is reported as a dominant mechanism of production of sulfate. A numerical study by Barth et al. (2000) reports that for the in-cloud conditions, aqueous phase reactions accounts for 81% of sulfate production rate. According to their study total of $\sim 50\% - 60\%$ of sulfate burden in the troposphere is produced by aqueous phase chemistry. Noteworthy, sulfate is a common component of aerosol particles (10%-67% of the

submicron particle mass is made of sulfate, 32% on average; see Zhang et al., 2007).

Sulfate aerosols cool Earth's climate by scattering sunlight and thus increasing Earth's shortwave albedo (direct radiative forcing) and by changing radiative properties of clouds (cloud albedo effect). According to the chapter 8 of IPCC Assessment Report (Myhre et al., 2013), the range of effective radiative forcings for all aerosol-radiation interactions is -0.95 to 0.05 W/m$^2$ and for aerosol-cloud interactions is -1.2 to 0.0 W/m$^2$. The level of scientific understanding in that report for the cloud albedo

effect is still marked as "low". From the air quality perspective, in extreme cases sulfur chemistry may lead to creation of acid rain or acid fog (Dianwu et al., 1988; Wang et al., 2016). Based on analysis of 20 modeling studies, the review by Faloona (2009) marks wet deposition of aerosol sulfate, dry deposition of $SO_2$ and heterogeneous (aqueous phase) oxidation of $SO_2$ in aerosol particles and clouds as the most challenging to quantify in models. For an overview of the representation of sulfur oxidation in regional and global models see Ervens (2015).

From the cloud microphysics stand point, aqueous phase oxidation of sulfur is interesting because it affects the CCN within water drops. The reaction is irreversible, meaning that the produced sulfate remains within the water drops and increases the dissolved CCN mass. Collisions and the subsequent coalescence of water drops is another in-cloud irreversible process that affects the CCN sizes. As the water drops collide and coalesce, the created new water drop carries within the combined CCN mass

of all of its colliding predecessors. Efficient collisions between cloud droplets may quickly lead to the onset of precipitation which can in turn effectively cleanse the atmosphere from aerosol particles and water soluble trace gases. In non-precipitating clouds, aerosol particles that served as CCN are altered by cloud microphysical and chemical processes and then returned to the atmosphere after water drops evaporate. The cloud-processed aerosol particles can be observed in measurements (Hoppel et al., 1986, 1994), (Werner et al., 2014), (Hudson et al., 2015). The case without precipitation is especially interesting as it

allows for aerosol-cloud interactions to loop for several cloud life- cycles without removing the altered aerosol particles. The



cloud-processed aerosol particles may again serve as CCN and influence microphysical properties of the next generation of clouds. The study of Pruppacher and Jaenicke (1995) estimates that on global average an atmospheric aerosol particle has been cycled 3 times by cloud systems. The LMC scheme introduced here offers a chance to represent the effects of such cloud-processing on CCN sizes stemming from both collisions between water drops and aqueous phase oxidation reaction within

water drops. To the authors knowledge, the presented scheme is the first to represent the impact of both collisions and aqueous phase chemistry on the aerosol size spectrum in the Lagrangian microphysics framework.

The structure of the presented work is as follows: Section 2 presents briefly the Lagrangian scheme available in *libcloudph++*. Section 3 discusses the design and user interface of the new aqueous chemistry scheme. Section 4 presents the

validation tests of the new scheme. Section 5 discusses the results from simulations where the LMC scheme is incorporated into a simple model of a stratocumulus cloud. The effects of both collisions between water drops and aqueous phase oxidation of sulfur on the aerosol particle size distribution are presented.

## 2   Lagrangian microphysics scheme

The Lagrangian scheme used in this work is described in detail in Arabas et al. (2015) and this section only briefly summarizes

its major concepts. In the Lagrangian approach to modeling cloud microphysics the computational domain is filled with "numerical point particles" representing a multiplicity of real particles (aerosol particles, cloud droplets or rain drops) of the same properties. Following the nomenclature introduced by Shima et al. (2009), the "numerical particles" are labeled here as superdroplets (SD). Each SD has a set of attributes describing the properties of the aerosol particles or water drops it represents. As discussed in Arabas et al. (2015), for microphysical purposes, the required attributes are: multiplicity ($\mathcal{N}$), the position of

SD in the computational domain, the wet radius ($r_w$), the dry radius ($r_d$)[2] and the hygroscopicity parameter ($\kappa$)[3]. The aqueous phase chemistry scheme extends the list of required attributes by masses of chemical compounds dissolved in droplets, see Sec. 3.

The key attribute of the Lagrangian microphysics scheme is the SD multiplicity. Multiplicity defines the number of aerosol particles or water drops represented by a given SD. All particles represented by one SD are assumed to be identical. Using

multiplicites allows to reduce the complexity of the problem and enables efficient numerical computations.

The Lagrangian scheme used here requires no division into artificial categories of aerosol particles, cloud and rain water, as it is often done in the bulk schemes, for example Kessler (1995), Seifert and Beheng (2001), Morrison and Grabowski (2007). All the modeled microphysical processes are represented by calculating the changes to the SD attributes. The equation of condensational growth is solved iteratively for each SDs wet radius (see Sec. 5.1.3 in Arabas et al. (2015) for details).

The process of condensational growth from deliquescent aerosol particles to cloud droplets is thus resolved and no additional parametrisation of cloud droplet activation is required, as it is again often done in the bulk microphysics schemes (for example

---

[2] It is a volume equivalent radius for solute in the water drop.

[3] Following Petters and Kreidenweis (2007) it is a single parameter representing the hygroscopicity of the solvent.





Morrison and Grabowski (2007)). Following Shima et al. (2009) the collisions between SDs are represented using Monte-Carlo scheme (see Sec. 5.1.4 in Arabas et al. (2015) for details).

Information about SD attributes is retained within the model throughout the whole simulation. This means that the size distribution of both water drops and aerosol particles in each computational grid-cell can be easily obtained by taking into

account the SD attributes of $r_w$, $r_d$ and $\mathcal{N}$. As a result, the Lagrangian scheme is capable of resolving the changes to both aerosol and water drop size distributions. The same functionality is offered by the 2-dimensional bin schemes, for example Ovchinnikov and Easter (2010) or Lebo and Seinfeld (2011). However, the Lagrangian approach greatly reduces the numerical diffusion errors. As discussed in Unterstrasser et al. (2016), it does introduce statistical errors, i.e. fluctuations between different realizations of the same collision/coalescence scenario. These errors are easier to minimize than diffusion numerical errors, for

example by increasing the number of SDs in the computational domain or by averaging over an ensemble of simulation runs. Lagrangian schemes solve ordinary differential equations instead of the partial differential equations encountered in the bin schemes, which is computationally more efficient.

The Lagrangian methods are becoming a well known tool for studying cloud microphysics in both warm-clouds (Shima et al. (2009), Arabas and Shima (2013), Arabas and Shima (2017), Andrejczuk et al. (2010), Andrejczuk et al. (2014), Hoffmann

(2017), Grabowski et al. (2018), Sardina et al. (2018)); warm-rain clouds (Riechelmann et al. (2012), Junghwa et al. (2014), Naumann and Seifert (2015)); and ice-phase clouds (Sölch and Kärcher (2010), Unterstrasser and Sölch (2014)). None of the above however, included description of the aqueous phase chemical reactions happening within cloud droplets.

## 3  Aqueous phase chemistry scheme

In order to represent the chemical composition of water drops the aqueous-phase chemistry scheme extends the list of SD attributes. The additional attributes are defined as the total mass of each of the chemical compounds in a given SD (including both the dissolved and, if applicable, dissociated fraction). An additional variable – the mass of the $H^+$ ions – is also added, in order to keep track of SD's pH. This results in eight new SD attributes needed for simulations with aqueous phase chemistry:

- the total mass of dissolved $O_3$,

- the total mass of dissolved $H_2O_2$,

- the total mass of dissolved $SO_2$ (including: $SO_2 * H_2O$, $HSO_3^-$ and $SO_3^{2-}$),

- the total mass of dissolved $CO_2$ (including: $CO_2 * H_2O$, $HCO_3-$ and $CO_3^{2-}$),

- the total mass of dissolved $NH_3$ (including: $NH_3 * H_2O$ and $NH_4^+$),

- the total mass of dissolved $HNO_3$ (including: $HNO_3(aq)$ and $NO_3^-$),

- the total mass of created $H_2SO_4$ (including: $HSO_4^-$ and $SO_4^{2-}$)

- the total mass of $H^+$ ions.





The scheme needs to be coupled to the driver model that provides information about the environment in which SD are immersed (i.e. temperature, humidity, trace gas mixing ratio, and wind speed). The representation of aqueous phase chemistry more than doubles the number of required SD attributes and significantly increases the computational time. On the other hand, thanks to the added attributes, the mass of any ion for any SD can be easily diagnosed using just a dissociation constant. This, in

turn, allows for a very straightforward representation of the aqueous chemical processes and does not call for any additional parametrisation.

All aqueous-phase chemistry included in the scheme is formulated under the assumption that solution droplets are diluted. Therefore, in the LMC scheme, chemical processes are only performed for SDs with ionic strength smaller than 0.02 moles/liter (the same criterion is used for example in Ovchinnikov and Easter, 2010). In practice, this condition results in excluding from

aqueous chemistry calculations SDs with small wet radii (i.e. SDs representing haze particles and very small cloud droplets). Exclusion of SDs with small wet radii also prevents numerical issues during condensation procedure when changes in dry radius caused by oxidation could prevent convergence of the condensation scheme during the initial rapid growth of cloud droplets during activation.

Combining the Lagrangian microphysics scheme with aqueous phase chemistry is straightforward. Condensation/evaporation

does not affect the chemical attributes of SDs. During collisions the mass of chemical compounds is summed when recalculating SD attributes (it is an extensive parameter). In principle the $\kappa$ attribute should be recalculated in every time-step based on the new chemical composition of each SD. However, the $\kappa$ values relevant for this study are very similar - the $\kappa$ value of ammonium bisulfate is 0.61 (Petters and Kreidenweis, 2007) and of sulfuric acid is 0.64 (Kim et al., 2016). Therefore, the hygroscopicity parameter is assumed to be constant.

As discussed in Kreidenweis et al. (2003), one of the major sources of errors in modeling the in-cloud aqueous phase oxidation are the uncertainties when resolving the cloud droplet size distribution. It follows that combining the aqueous phase chemistry model with the very detailed Lagrangian microphysics scheme is beneficial. This will be discussed further in Sec. 4.

### 3.1    Dissociation

Dissociation is a reversible process of splitting of the molecules dissolved in water drops into ions. It is treated as an equilibrium

process and is described using the dissociation constants. The dissociation constant of chemical compound A is denoted here by $\mathbb{K}_A$. The dissociation constants are corrected for temperature using the formula of van 't Hoff (1885):

$$\mathbb{K}_A(T) = \mathbb{K}_A(T_0) \exp\left(\frac{-\Delta H_D}{R}\left(\frac{1}{T} - \frac{1}{T_0}\right)\right), \tag{1}$$

where $\Delta H_D$ denotes the reaction enthalpy of dissociation at constant temperature and pressure. The list of considered dissociation constants as well as their temperature correction coefficients is available in Tab. C2. The dissociation of water, although

very small, is also taken into account[4]. No temperature correction is applied to water dissociation constant.

---

[4] The concentration of undissociated water molecules is so big that it is usually assumed constant and it traditionally multiplies the dissociation constant of water. This leads to a different definition of the dissociation constant for water: $\mathbb{K}_{H2O} = [H^+][OH^-]$



It is assumed that there is no electric charge of water drops and therefore the concentrations of positive and negative ions created during dissociation should balance each other. Using dissociation constants (see Tab. C2), all ion concentrations can be expressed as a function of the total concentration of the dissolved chemical compounds and the concentration of H⁺ ions. The neutral charge condition can be expressed as

$$
[\text{H}^+] + \overbrace{\frac{[\text{N}^{-\text{III}}][\text{H}^+]\mathbb{K}_{\text{NH3}}}{\mathbb{K}_{\text{H2O}} + \mathbb{K}_{\text{NH3}}[\text{H}^+]}}^{\text{positive ions}} = \frac{[\text{N}^{\text{V}}]\mathbb{K}_{\text{HNO3}}}{[\text{H}^+] + \mathbb{K}_{\text{HNO3}}} + \frac{[\text{S}^{\text{IV}}]\mathbb{K}_{\text{SO2}}([\text{H}^+] + 2\mathbb{K}_{\text{HSO3}})}{([\text{H}^+]^2 + [\text{H}^+]\mathbb{K}_{\text{SO2}} + \mathbb{K}_{\text{SO2}}\mathbb{K}_{\text{HSO3}})} +
$$

$$
\underbrace{\frac{\mathbb{K}_{\text{H2O}}}{[\text{H}^+]} + \frac{[\text{S}^{\text{VI}}]([\text{H}^+] + 2\mathbb{K}_{\text{H2SO4}})}{([\text{H}^+] + \mathbb{K}_{\text{H2SO4}})} + \frac{[\text{C}^{\text{IV}}]\mathbb{K}_{\text{CO2}}([\text{H}^+] + 2\mathbb{K}_{\text{HCO3}})}{([\text{H}^+]^2 + [\text{H}^+]\mathbb{K}_{\text{CO2}} + \mathbb{K}_{\text{CO2}}\mathbb{K}_{\text{HCO3}})}}_{\text{negative ions}} . \tag{2}
$$

The [ ] brackets denote the concentration of each of the chemical compounds, (traditionally defined in units of moles per liter), capital letters denote chemical compound and roman numbers mark its oxidation state. In Eq. (2) the dissociation constants of $\text{SO}_3^{2-}$, $\text{CO}_3^{2-}$, and $\text{SO}_4^{2-}$ ions (i.e. $\mathbb{K}_{\text{HSO3}}$, $\mathbb{K}_{\text{HCO3}}$, and $\mathbb{K}_{\text{H2SO4}}$) are multiplied by a factor of two, to take into account bigger electric charge number of those ions.

Equation. (2) has only one unknown variable – the new equilibrium concentration of the H⁺ ions. The new concentration is obtained iteratively using bisection algorithm[5]. The algorithm searches for solution between pH = -1 and pH = 9. The lower bound for the pH scale is unrealistically acidic and is only necessary at the start of the simulation when the initial SDs have very small volume and thus highly acidic pH. The upper bound is set arbitrarily, but is sufficient for the expected pH of the modeled droplets. At the end of the dissociation procedure the mass of H⁺ ions is updated based on the new equilibrium concentration.

When the SD wet radius is quickly changing, for example during the initial condensational growth of cloud droplet or rain drop evaporation, the dissociation procedure requires small time-steps to reach convergence. The time-step used in dissociation procedure can be divided into user-specified number of sub-steps in order to prevent limiting the overall simulation time-step by dissociation.

## 3.2 Dissolution

The amount of the chemical compound that can dissolve into water drop from the gas phase is proportional to its partial pressure above the surface of the drop. Due to the longer timescale of the process, in contrast to dissociation, the transfer between the gas and liquid phase is not treated as an instantaneous process. Assuming that the water drop is internally mixed, the gas-liquid transfer is limited by the diffusion of gas phase particles to the drop surface (gas-phase limitation) and the probability that the molecule will enter the drop after collision (interfacial limitation). Following chapter 8.4.2 in Warneck (1999), for a chemical compound "A" the rate of transfer from the gas phase to the aqueous phase equals

$$
\frac{d[\text{A}]}{dt} = \left( \frac{4r_w}{3\langle v \rangle \alpha_{M_A}} + \frac{r_w{}^2}{3D_A} \right)^{-1} \left( c_{A\infty} - \frac{[\text{A}]}{\mathbb{H}_A^{eff} RT} \right) \tag{3}
$$

[5] TOMS 748 algorithm from *Boost* library. See www.boost.org/doc for documentation and Alefeld et al. (1995) for derivation





where $D_A$ and $\alpha_{M_A}$ are the diffusion and mass accommodation coefficients for the chemical compound "A", $<v> = \sqrt{\frac{8RT}{\pi M_A}}$ is the average velocity of the molecules calculated from the Maxwell-Boltzmann distribution function, $M_A$ is the molar mass of the chemical compound "A", $c_{A\infty}$ is the ambient concentration of the trace gas "A" and $\mathbb{H}_A^{eff}$ is the effective equilibrium constant for dissolution of the chemical compound "A". The dissolution constants depend on the temperature following similar

relation as for dissociation Eq. (1). Table C3 shows the equilibrium dissolution constants and their temperature corrections and Tab. C4 presents the diffusion and mass accommodation coefficients. The term "effective" marks that the dissolution constants take into account the increase of the efficiency due to dissociation (see Seinfeld and Pandis (1998) chap. 7.4 for the exact equations). Equation (3) is solved for each SD and for each of the considered trace gases. It is solved implicitly with respect to aqueous-phase concentration and explicitly with respect to the gas-phase concentration. The input ambient trace gas

concentration is calculated from the trace gas mixing ratio provided by the driver model to which the LMC scheme is coupled. Obtained aqueous phase concentration is recalculated to the mass of dissolved chemical compounds and the corresponding SD attribute is updated. The changes in the ambient trace gas mixing ratios are calculated by LMC scheme by summing the changes in chemical composition in all SDs in a given grid-cell and then subtracting them from the trace gas mixing ratio of the driver model. This approach does not ensure that per each time-step the total dissolved mass of a given trace gas does not

exceed the available ambient mixing ratio. To prevent that, relatively short time-steps should be applied. If necessary the user can divide the model time-step into sub-steps.

### 3.3 Oxidation

The reaction rates of oxidation by ozone and hydrogen peroxide can be described as (Hoffmann and Calvert, 1985):

$$\mathbb{R}_{O3} = \left.\frac{d[S^{VI}]}{dt}\right|_{O3} = \left(k_0 + \frac{k_1 \mathbb{K}_{SO2}}{[H^+]} + \frac{k_2 \mathbb{K}_{SO2} \mathbb{K}_{HSO3}}{[H^+]^2}\right)[O_3][SO_2 * H_2O] \tag{4}$$

$$\mathbb{R}_{H2O2} = \left.\frac{d[S^{VI}]}{dt}\right|_{H2O2} = \frac{k_3 \mathbb{K}_{SO2}}{1 + k_4[H^+]}[H_2O_2][SO_2 * H_2O] \tag{5}$$

where $\mathbb{R}_A$ is the reaction rate of the chemical compound "A" and $k_{0,...,4}$ are the reaction rate coefficients. $k_{0,...,4}$ depend on the temperature following similar relation as for dissociation Eq. (1). Table C5 shows the values of reaction rate coefficients and their temperature correction coefficients.

Equations (4) and (5) return the new concentration of $H_2SO_4$ created in each SD in each time-setp. Based on the new concentration, the new mass of $H_2SO_4$ and the new dry radius are calculated and the corresponding SD attributes are updated. The dry particle density of 1.8 g/cm$^3$ is assumed while evaluating the dry radius from the $H_2SO_4$ mass.

For the typical atmospheric conditions, say pH between 3 and 6 (i.e. [H$^+$] between $10^{-3}$ and $10^{-6}$), it can be said that the

rate of oxidation by $H_2O_2$ does not depend on pH (see Tab. C2 for the dissociation constant values). In contrast, oxidation by ozone depends strongly on pH of the solution and can become very fast if pH is high. For example, increasing pH by 1 point results in approximately 100 increase in $O_3$ reaction rate.



## 3.4 Initialization

The initial aerosol is assumed to be ammonium bisulfate ($NH_4HSO_4$), with dry particle density of 1.8 g cm$^{-3}$. Using dry particle density and dry radius of each SD, the initial mass of $H^+$, $NH^+$ and $SO_4^{2-}$ ions is calculated. The initial mass of other molecules and ions is equal zero and is therefore not in equilibrium with the initial ambient trace gas conditions. For the

initial conditions above supersaturation it is advisable to allow for a spin-up period with only condensation/evaporation and the equilibrium chemical processes enabled, to allow the model to reach equilibrium.

## 4 Validation

The LMC scheme is set to reproduce results from model intercomparison study by Kreidenweis et al. (2003), where several bulk and bin schemes representing cloud microphysics and aqueous-phase chemistry were tested in an adiabatic parcel model

setup. A parcel model is a 0-dimensional model that represents an idealized scenario of a finite volume of air rising adiabatically with a constant vertical velocity. As the parcel of air raises, its temperature decreases leading to supersaturation. This results in activation and further condensational growth of cloud droplets. For the studied oxidation reaction, the presence of liquid water enables aqueous-phase chemical reactions and leads to creation of sulfuric acid within cloud droplets. The collisions between cloud droplets are not included in the parcel simulations to allow an easy comparison with Kreidenweis et al. (2003).

The initial conditions are the same as in Kreidenweis et al. (2003) and are provided for convenience in Tab. 1. The simulation starts below cloud base (i.e. with subsaturation). The initial aerosol is ammonium bisulfate and the initial aerosol particle size distribution is assumed to be lognormal with one mode

$$n(r_d) = \frac{n_{tot}}{r_d \sqrt{2\pi} ln(\sigma_g)} exp\left(-\frac{(ln(r_d) - ln(\overline{r_d}))^2}{2ln^2(\sigma_g)}\right) \tag{6}$$

where $n(r_d)$ is the spectral density function of aerosol particle sizes, $n_{tot}$ is the total aerosol concentration, $\overline{r_d}$ is median radius

and $\sigma_g$ is the geometric standard deviation.

The parcel model employed in this study uses dry air density $\rho_d$, dry air potential temperature $\theta$, water vapor mixing ratio $r_v$, and mixing ratios of ambient trace gases as model variables. In order to calculate $\rho_d$ at each time-level (or each height-level of the parcel ascent) the model assumes a vertical profile of pressure. In the presented simulations the assumed pressure profile is obtained by integrating the hydrostatic equation and assuming that density of air is constant at each height-level (piecewise

constant). Then, at each level, $\rho_d$ is calculated from the ideal gas law taking into account the current $r_v$ and $\theta$. Because the simulated air parcel is assumed to be adiabatic, only processes resolved by the Lagrangian scheme can change $\theta$, $r_v$ and other trace gas mixing ratios. In each model time-step, the Lagrangian microphysics scheme changes $\theta$ and $r_v$ according to Eq. 25 and 26 from Arabas et al. (2015). The changes in the trace gas mixing ratios are resolved following procedure discussed in Sec. 3.2. It is assumed that the initial mass of dry air within the parcel is 1 kg.

Figure 1 shows the general physical and chemical conditions from the cloud base up to the end of the test run 1.2 km above the cloud base. Two vertical axes are used, representing either the time or the height above the cloud base. Figure 1a shows the liquid water mixing ratio (LWC). The increase in LWC is linear and the LWC reaches above 2 g/kg at height 1.2km



**Table 1.** Initial conditions for the adiabatic parcel test.

| factor | value | units |
|---|---|---|
| number of super-droplets | 1024 | - |
| time-step | 0.1 | s |
| | | |
| temperature at t = 0 | 285.5 | K |
| pressure at t = 0 | 950 | hPa |
| relative humidity at t = 0 | 95 | % |
| updraft velocity | 0.5 | m/s |
| | | |
| median radius | 0.4 | $\mu$m |
| geometric standard deviation | 2 | - |
| total aerosol number concentration | 566 | cm$^{-3}$ |
| dry particle density | 1.8 | g/cm$^3$ |
| hygroscopicity | 0.61 | - |
| | | |
| concentration of $SO_2$ at t = 0 | 0.2 | ppb-v |
| concentration of $O_3$ at t = 0 | 50 | ppb-v |
| concentration of $H_2O_2$ at t = 0 | 0.5 | ppb-v |
| concentration of $CO_2$ at t = 0 | 360 | ppm-v |
| concentration of $HNO_3$ at t = 0 | 0.1 | ppb-v |
| concentration of $NH_3$ at t = 0 | 0.1 | ppb-v |

above the cloud base. Figure 1b shows the total $SO_2$ concentration (both in gas phase and dissolved in water) in ppb units. The concentration of $SO_2$ is decreasing due to oxidation taking place in cloud droplets. Figure 1c shows the water weighted average pH of cloud droplets. The pH near the cloud base is very low due to acidic nature of the assumed initial aerosol and small size of activated cloud droplets. As the drops grow bigger and become more diluted, the average pH increases. Figure 1 compares well

5   with Fig. 1 from Kreidenweis et al. (2003), the overall differences between the two figures are small. The biggest discrepancy is in the LWC profile. This is arguably caused by different pressure profiles in the two parcel models (the pressure profile in Kreidenweis et al. (2003) is not specified).

At the end of the test simulation, 84% of $SO_2$ is converted into $H_2SO_4$ and the final water weighted average pH is equal to 4.83. The total sulfate production is 168 ppt with 100 ppt produced by the $H_2O_2$ reaction path and 68 ppt produced by the $O_3$

10   reaction path. Based on Fig. 2 in Kreidenweis et al. (2003), the range of average pH values reported by different size resolving (bin) schemes was between 4.82 and 4.85, and the range of total sulfate production values was between 170 – 180 ppt. Based on Fig. 3 in Kreidenweis et al. (2003) the production by $H_2O_2$ ranged between 85 and 105 ppt, and by $O_3$ between 70 and 85 ppt for the size resolving schemes. In short, the results from the Lagrangian scheme are close to the range of values reported





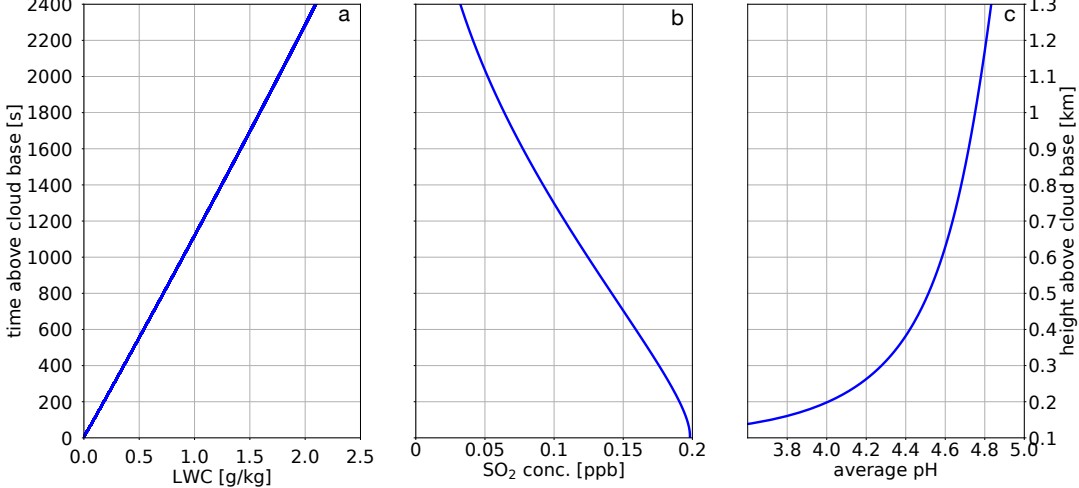

**Figure 1.** Physical and chemical conditions in the adiabatic parcel model. Panel a shows the liquid water mixing ratio (LWC), panel b shows the $SO_2$ concentration (both gas phase and dissolved) panel c shows the water weighted average pH of the simulated population of water drops.

by bin schemes.

Microphysics schemes taking part in the Kreidenweis et al. (2003) intercomparison study reported significant differences between the number of activated cloud droplets. Based on Tab. 1 in Kreidenweis et al. (2003) the droplet number concentration
at the cloud base varied between 275 and 358 cm$^{-3}$. One of the differences between the bin schemes responsible for causing this discrepancy is the different water vapor mass accommodation coefficient $\alpha_M$ leading to different predicted maximum supersaturation. Figure 6 in Kreidenweis et al. (2003) shows that the observed maximum supersaturations were lower (between 0.23% − 0.26%) for schemes using high values of $\alpha_M$ (either 0.5 or 1). In contrast, a scheme using $\alpha_M = 0.042$ predicted maximum supersaturation equal to 0.37%. The Lagrangian scheme used in this study reports concentration 264 cm$^{-3}$ at RH =
1 and 272 cm$^{-3}$ at the level of maximum supersaturation. The maximum supersaturation is equal to 0.27%. The Lagrangian scheme assumes $\alpha_M$ equal to unity and therefore fits with the trend of high $\alpha_M$ causing lower supersaturation presented in Fig. 6 in Kreidenweis et al. (2003).

Another cause for the discrepancy between the bin schemes listed in the intercomparison study are the different sizes and locations of bins in different models. Along those lines, here it is tested how sensitive the Lagrangian scheme is to the number
of SDs. The results of this test are summarized in Fig. 2 showing the cloud droplet concentration at the cloud base (a), the maximum supersaturation (b), the average pH (c) and the total sulfate production (d). The results are plotted against the logarithm of base two of the number of SDs in the computational domain (meaning that "0" represents one SD and "10" represents 1024 SDs). All values seem to converge for SD numbers greater than 128. The average pH, maximum supersaturation and total sulfate production do not change for those four test-runs. The concentration of droplets at the cloud base varies

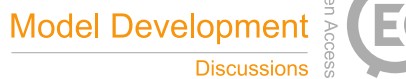



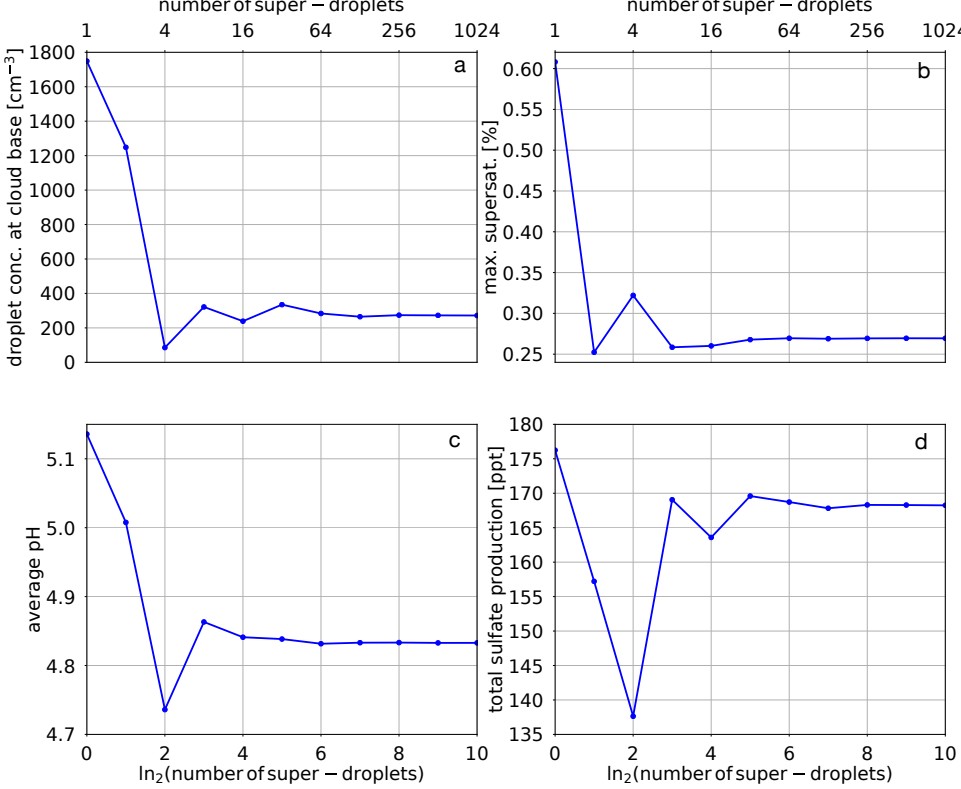

**Figure 2.** Results of the convergence test for the adiabatic parcel simulations. All figures show how a given parameter depends on the number of SDs (shown on the abscissa as the logarithm of base two of the number of SDs). Panel a the cloud droplet concentration at the cloud base, panel b the maximum supersaturation, panel c the water weighted average pH at the end of simulation and panel d the total sulfate production.

little (between $265\,\mathrm{cm}^{-3}$ and $274\,\mathrm{cm}^{-3}$). The cloud droplet concentration from simulation with 128 SDs is the largest outlier. The concentrations from simulations with SD number between 256 and 1024 vary between $274\,\mathrm{cm}^{-3}$ and $272\,\mathrm{cm}^{-3}$. For SD numbers between 32 and 128 there are insignificant changes in the maximum supersaturation. The values of pH vary by 0.01 and the total sulfate production increases by 1 ppt. There are, however, large differences between the number of droplets at the cloud base (between $265\,\mathrm{cm}^{-3}$ and $335\,\mathrm{cm}^{-3}$). This confirms the observations from Kreidenweis et al. (2003) that the predicted cloud droplet number concentration strongly depends on the representation of the size distribution of modeled aerosol particles and cloud droplets and that this may become a major source of uncertainties in the microphysics representation. Decrease in the SD number below 32 leads to a big variance in the cloud droplet concentration as well as other parameters.

Figure 3 shows the simulated modification of the aerosol size distribution. Red line depicts the initial distribution and the green line shows model state at the end of adiabatic parcel test. For convenience, Fig. 3 uses both logarithmic (left panel) and linear (right panel) scale on ordinate. The change in aerosol size distribution is caused by oxidation. Aerosol particles that are too small to become cloud droplets are not affected by aqueous phase oxidation and they do not grow in size. Large aerosol





particles grow in size due to $H_2SO_4$ production during oxidation, but the increase in size is small compared to their initial size. The smallest activated aerosol particles are those that are affected most by oxidation. The increase in their size due to the produced $H_2SO_4$ is the largest compared to their initial size. In short, oxidation produces a "gap", often labeled the "Hoppel minimum", between the CCN processed by the cloud and the smaller unactivated aerosol particles.

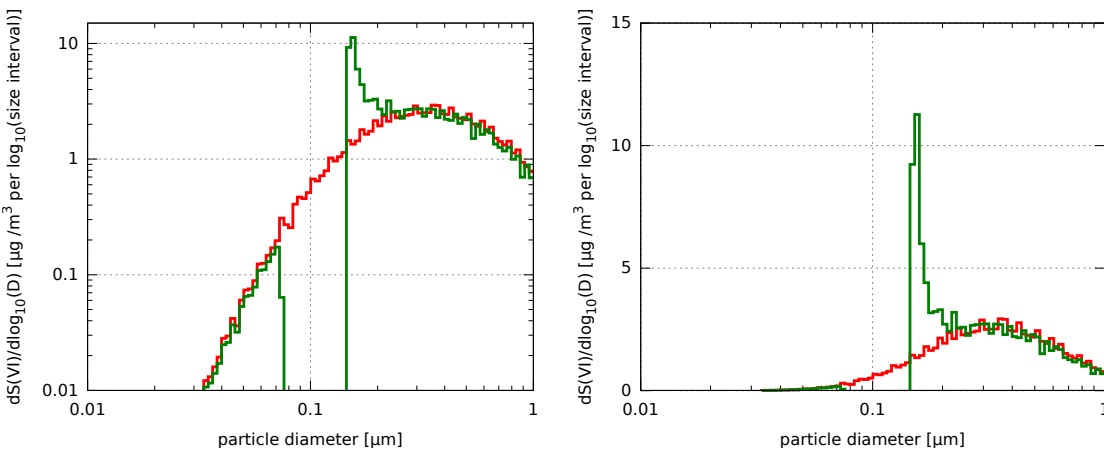

**Figure 3.** Modification of the dry aerosol sulfate mass. Red line shows the initial condition and green line shows the final model state. Left panel uses logarithmic ordinate and right panel uses linear ordinate.

The effect of in-cloud sulfate production on aerosol particle size distribution presented in Fig. 3, combined with other tests presented in this chapter, document the correctness of the implementation of aqueous chemistry in the Lagrangian scheme. The formation of the "Hoppel minimum" was reported by many observational studies, see Hoppel et al. (1994); Bower et al. (1997); Hudson et al. (2015). Figure 3 compares well with aerosol size distribution plots from the intercomparison study shown in Fig. 9 in Kreidenweis et al. (2003). Other numerical schemes also reported the formation of "Hoppel minimum", see for
example Flossmann (1994); Feingold and Kreidenweis (2000, 2002); Ovchinnikov and Easter (2010).

## 5    Example simulations

### 5.1    2D kinematic model

The kinematic model mimics a single 2D eddy spanning a stratocumulus cloud deck and a boundary layer below it. The model is based on a test scenario from the 8[th] International Cloud Modeling Workshop (ICMW; Muhlbauer et al., 2013, case 1). The
velocity field is prescribed as in Szumowski et al. (1998); Morrison and Grabowski (2007); Rasinski et al. (2011). The same model was used when presenting the initial release of *libcloudph++*, see Sec. 2 in Arabas et al. (2015) for the details of the model formulation. The model operates on the Eulerian grid. At each model time-step, the temperature, moisture, and trace gas fields are advected using the prescribed velocity field. Then, the model variables are passed to the LMC scheme, where the microphysical and chemical processes are resolved. Finally, the source and sink terms due to microphysics and chemistry are



calculated and applied in each model grid-cell as described in Sec. 2 and 3.

The collisions between water drops are represented using the geometric kernel with collision efficiency for big drops from Hall (1980) and for small droplets from Pinsky et al. (2008). For big drops, the collision efficiencies were obtained from the

fit to measurements, see Hall (1980). For small droplets, the collision efficiencies were based on numerical simulations taking into account turbulence typical for stratocumulus clouds, see Pinsky et al. (2008). The collision efficiencies are provided via a look-up tables for different drop sizes.

The initial conditions are summarized in Tab. 2. The computational domain size is 1.5 km in both directions and the computational grid is composed of 75×75 cells of equal size (the grid lengths are 20 m) and is periodic in the horizontal direction.

The initial air density profile corresponds to the hydrostatic equilibrium with the pressure of 1015 hPa at the bottom of the domain. At the beginning of the simulation it is assumed that there is no condensed water, and the initial profiles of $\theta$ and $r_v$ are constant with altitude. To keep the simulation setup simple and due to a relatively low vertical extent of the computational domain, the initial trace gas volume fractions are also assumed to be constant with altitude. This unrealistic initial condition results in very high initial supersaturation in the upper part of the domain. As a consequence a $10^5$ second ($\sim$ 2h 45min) spin-up

period is necessary to allow for the simulated water drops to reach equilibrium with their environment. During the spin-up only the reversible processes (condensation and evaporation, dissolving of trace gases and dissociation into ions) are allowed and the supersaturation is limited to 5% (RH=1.05). After spin-up the simulations are run for 30 minutes. The chosen simulation time is enough to deplete the $SO_2$ available in the cloudy part of the domain as well as to create precipitation.

Similarly to the adiabatic parcel test, the initial aerosol is ammonium bisulfate and the aerosol particle size distribution is

lognormal with one mode. The initial condition for trace gases is defined in terms of volume fractions and then translated to mixing ratios that serve as the the model variables. The initial $SO_2$, $O_3$ and $H_2O_2$ volume fractions are taken from the simulation setup used in Ovchinnikov and Easter (2010). The values for $SO_2$ and $O_3$ are based on the measurements from MASE campaign (Wang et al., 2008) and the value for $H_2O_2$ is based on the representative values for the Eastern Pacific Ocean (Genfa et al., 1999). The $NH_3$, $HNO_3$ and $CO_2$ volume fractions are the same as in the parcel test from Sec. 4.

The setup detailed in Tab. 2 corresponds to "very clean conditions" (i.e. low aerosol particle concentrations). Initial aerosol particle sizes are also relatively small. Three additional simulation cases are studied to check the sensitivity of the model to different conditions. In case1 the reversible chemical processes are allowed, but oxidation is prohibited. In case2 the initial volume fraction of $NH_3$ is increased and in case3 the initial aerosol size distribution is changed. The conditions for all the sensitivity simulation cases are summarized in Tab. 3.

As discussed in Flossmann (1994), the initial chemical scenario is idealized. For instance, although the initial conditions represent clean maritime environment, the setup lacks sea salt aerosol particles. As discussed by Twohy et al. (1989), sea salt aerosol particles are alkaline, which may in turn increase the pH of water drops and thus affect the oxidation rate. On the other hand, a study by von Glasow and Sander (2001) indicates that alkaline sea salt particles are quickly converted to acidic due to the uptake of HCl vapor. More importantly, including sea salt would result in aerosol particles with very different

hygroscopicity values ($\kappa$ of ammonium bisulfate is 0.61, whereas $\kappa$ of NaCl is 1.28; Petters and Kreidenweis, 2007). Including





**Table 2.** Initial conditions for base case of 2-dimensional kinematic model.

| factor | value | units |
|---|---|---|
| number of super-droplets | 256 | #/grid-cell |
| model time-step | 1 | s |
| Lagrangian scheme time-step | 0.1 | s |
| dry air potential temperature at t = 0 | 289 | K |
| water vapor mixing ratio at t = 0 | 7.5 | g/kg |
| pressure at z = 0 | 1015 | hPa |
| | | |
| median radius | 0.05 | $\mu$m |
| geometric standard deviation | 1.8 | - |
| total aerosol number concentration | 50 | cm$^{-3}$ |
| dry particle density | 1.8 | g/cm$^3$ |
| hygroscopicity | 0.61 | - |
| | | |
| concentration of $SO_2$ at t=0 | 0.2 | ppb-v |
| concentration of $O_3$ at t=0 | 25 | ppb-v |
| concentration of $H_2O_2$ at t=0 | 0.4 | ppb-v |
| concentration of $CO_2$ at t=0 | 360 | ppm-v |
| concentration of $HNO_3$ at t=0 | 0.1 | ppb-v |
| concentration of $NH_3$ at t=0 | 0.1 | ppb-v |

**Table 3.** Initial conditions for sensitivity test cases of 2-dimensional kinematic model. Specified are: aqueous phase chemistry choice, initial volume fraction of $NH_3$, mean radius of the assumed lognormal aerosol particle size distribution $\overline{r_d}$, total aerosol concentration $n_{tot}$ and geometric standard deviation $\sigma_g$. Other parameters for each case are the same as in base case (Tab. 2) The parameters that distinguish each sensitivity test case are marked in bold.

| case | oxidation reaction | $NH_3$ [ppb-v] | $\overline{r_d}$ [$\mu$m] | $n_{tot}$ [cm$^{-3}$] | $\sigma_g$ |
|---|---|---|---|---|---|
| case1 | **off** | 0.1 | 0.05 | 50 | 1.8 |
| case2 | on | **0.4** | 0.05 | 50 | 1.8 |
| case3 | on | 0.1 | 0.05 | **150** | 1.8 |

sea salt would also result in the initial bi-modal size distribution with one mode representing smaller ammonium bisulfate aerosol particles and the second mode representing larger sea salt particles. In general, including sea salt should result in a very different condensational growth of aerosol particles. The setup used in this study also lacks other particles containing sulfate, such as ammonium sulfate or sulfuric acid aerosol particles. The reason behind the chosen setup, is that this is the first attempt





to include aqueous chemistry into the Lagrangian cloud microphysics and therefore the decision was made to start with the simplified setup.

The initial aerosol size distribution parameters are based on the test cases studied in Feingold and Kreidenweis (2002). The discussion presented in their study introduced two regimes for oxidation with regard to the mean aerosol size $\overline{r_d}$ and

precipitation: (i) for relatively small initial $\overline{r_d}$ production of sulfate enhances precipitation, (ii) for relatively big initial $\overline{r_d}$ production of sulfate suppresses precipitation. The overall impact depends strongly on the initial concentration of aerosol particles, see Feingold and Kreidenweis (2002) for the discussion. The short simulation time used in this study hinders analysis of the impact of oxidation on the overall precipitation. The work presented here focuses on the evolution of aerosol particle sizes and pH values of cloud and drizzle droplets. Future LES simulations should focus on the impacts of aqueous chemistry

on precipitation, cloud lifetime, and cloud dynamics.

The kinematic setup precludes any links between cloud microphysical processes and dynamics of the air motion. The setup limits the study to the smooth velocity and therefore smooth saturation fields and prevents mixing between air parcels with different trajectories and properties. On the other hand, the kinematic setup has low computational cost and allows easy testing and sensitivity analysis. Prescribing the velocity ensures that all changes to the aerosol particle and water drop size distribu-

tions are caused by the cloud microphysics and aqueous-phase chemistry alone. Moreover, the kinematic setup allows for a straightforward selection of the updraft and downdraft regions, further simplifying the analysis of the microphysical processes.

## 5.2    Results

Figure 4 shows the model state after 30 minutes of simulation from base case (see Tab. 2). Figure 4a shows the concentration of unactivated aerosol particles (defined as the SDs with wet radius smaller than 1 $\mu$m). The lower part of the plot (below 900 m)

shows cloud-free conditions and corresponds to the initial concentration of aerosol particles. The upper part of the plot shows the interstitial aerosol particles, i.e. those aerosol particles that did not activate. The difference between the upper and lower parts of Fig. 4a shows the impact of nucleation scavenging on aerosol population. The regions with slightly higher concentration of the in-cloud aerosol particles near the cloud base correspond to regions with low vertical velocities, lower supersaturations and thus lower concentrations of cloud droplets. Figure 4b shows the concentration of cloud droplets (defined as the SDs with

wet radii between 1 and 25 $\mu$m). It is nearly constant with height, that agrees with the observations in stratocumulus clouds (e.g. Pawlowska et al., 2000). The regions with lower cloud droplet concentrations correspond to the regions with drizzle (see Fig. 4f). Figure 4c shows the rain water mixing ratio (water drops with wet radius greater than 25 $\mu$m) using a logarithmic color scale. Rain forms quickly in the simulation due to the relatively high values of cloud droplet radii after the spin-up caused by the low initial aerosol particle concentration. The footprint of precipitation can be seen in Fig. 4a and f where the cloud droplet

concentration is depleted in regions of drizzle. Figure 4d shows the mean dry radius of all particles (both aerosol particles and water drops). The mean dry radius is increasing due to oxidation. In the updraft (left-hand side of panel d) new aerosol particles are advected into the cloudy region. Once the cloud droplets are formed, the aqueous phase oxidation starts to produce sulfate and changes the CCN size distribution. In the downdraft (right-hand side of panel d) cloud droplets are advected out of the cloud and they evaporate. The cloud-processed CCN are returned to the environment and change the ambient air aerosol





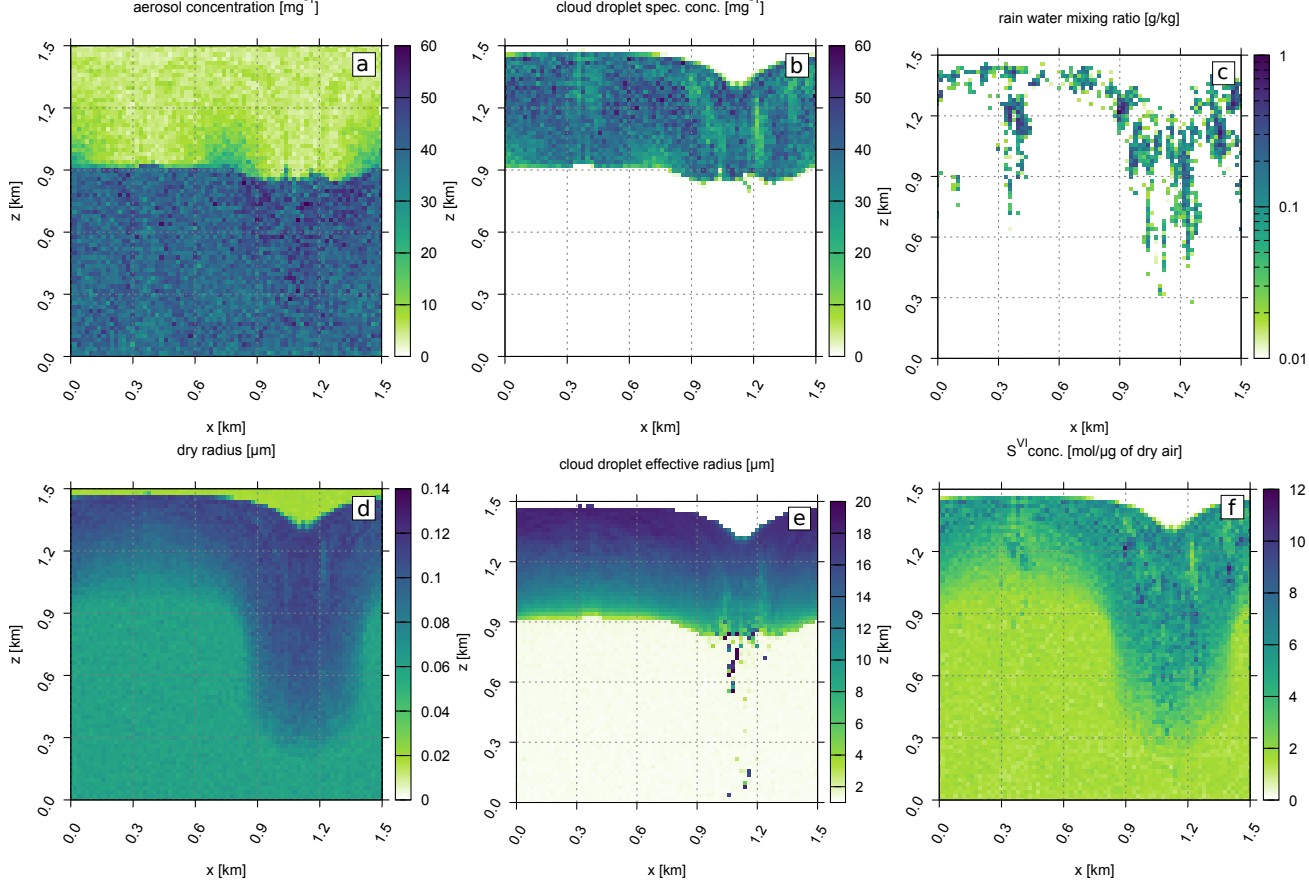

**Figure 4.** Base case setup (see Tab. 2). All panels depict model state after 30 minutes simulation time (excluding the spin-up) and show: aerosol concentration (a), cloud droplet concentration (b), rain water mixing ratio (c), mean dry radius (d), cloud droplet effective radius (e) and concentration of $S^{VI}$ molecules (f). The thresholds for particle radii are: aerosol < 1 $\mu$m; 1 $\mu$m < cloud < 25 $\mu$m; rain > 25 $\mu$m. Note the logarithmic scale for the rain water mixing ratio plot.

particle size distribution. Figure 4e depicts the cloud droplet effective radius. As expected, the effective radius increases with height. At the top of the cloud the effective radius reaches 20 $\mu$m, which is linked to the small cloud droplet concentration. High effective radii imply efficient drizzle production after the spin-up (usually water drop radius $\sim$ 12 $\mu$m is reported as the threshold value for efficient collisions between water drops and the production of precipitation, for example Rosenfeld

5  and Gutman (1994); Pawlowska and Brenguier (2003)). Figure 4f shows the concentration of $S^{VI}$ molecules (all molecules containing sulfur at +6 oxidation state) and represents molecules from the initial ammonium bisulfate ($NH_4HSO_4$) aerosol and the molecules created during oxidation. It corresponds to the mean dry radius plotted in Fig. 4d. Additionally, some effects of collisions and precipitation can be seen when comparing the irregular features from Fig. 4f with rain water mixing ratio in Fig. 4c. Precipitation displaces the largest water drops, which causes the irregular distribution of $S^{VI}$ molecules in cloudy





grid-cells. Figure 4f also shows that the Lagrangian scheme can track the dissolved chemical compounds in the evaporating rain drops below the cloud base.

Figure 4b, d, e and f show a layer of very clean air above the cloud which is caused by sedimentation of cloud droplets. In the downdraft region, the prescribed velocity field advects the clean layer into the domain. This feature is not present in the aerosol concentration plot (Fig. 4a) because the clean layer contains small aerosol particles with small sedimentation velocity. The depicted clean layer is an artifact caused by the prescribed velocity field and the absence of aerosol sources in the computational domain. The relatively short simulation time is chosen to minimizes the impact of the clean layer on the simulation.

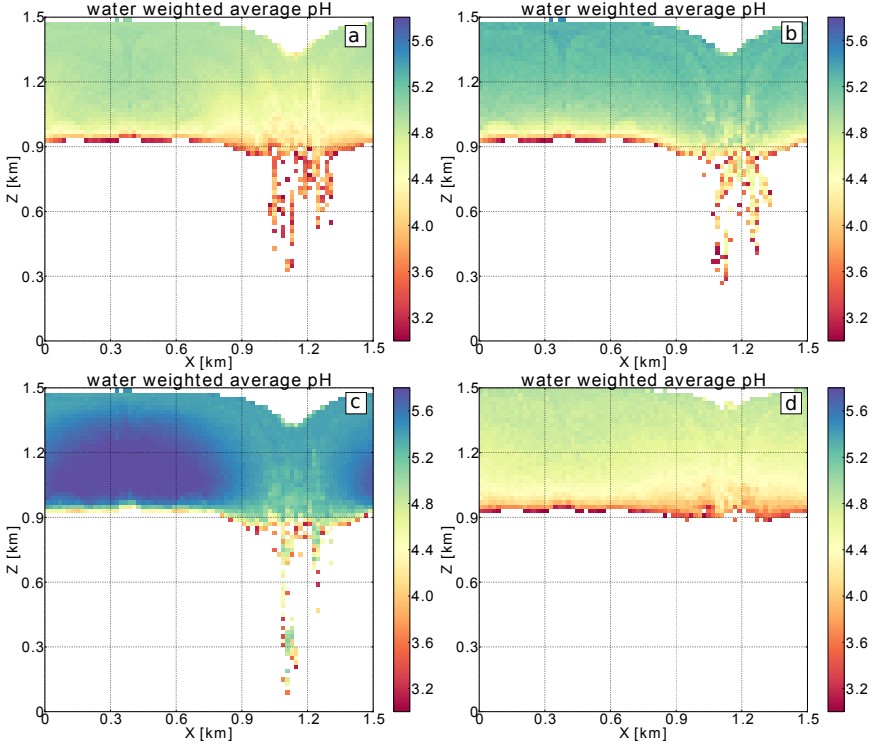

**Figure 5.** Liquid water weighted average pH from base case (a), case1 (b), case2 (c), case3 (d). See Tab. 2 and Tab. 3 for a definition of simulation setups.

Figure 5 shows the liquid water weighted average pH in each computational grid-cell from base case (a) and sensitivity test cases (b-d). In order to better adjust the color scale to the in-cloud pH variability, pH values below 3 that correspond to very acidic aerosol particles below the cloud base have been clipped. Figure 5 captures the pH of cloud droplets as well as the pH of some evaporating rain drops below the cloud base. The droplets in the downdraft of Fig.5a have higher acidity that is caused by $H_2SO_4$ created during aqueous phase oxidation. For base case, Fig. 5a, pH increases with height above the cloud base. Initially water drops are very acidic, but they grow in size and become more diluted. Even though $H_2SO_4$ is created during

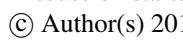



oxidation, the average pH still increases with height due to dilution. The same behavior is shown in the adiabatic parcel tests discussed in Sec. 4 and shown in Fig. 1. The increase of pH with height is also observed in 1-dimensional model representing processing of sulfur in small cumuli in marine environment (Alfonso and Raga, 2002). Due to the pH variability shown in Fig. 5a oxidation by $O_3$ happens mostly near the cloud top in base case. As discussed in Sec. 3.3, the rate of oxidation by

$O_3$ increases significantly with increasing pH (see Eq. 4) whereas oxidation by $H_2O_2$ does not depend on the acidity (Eq. 5). The study by Walcek and Taylor (1986) also reported that the pH of droplets increased with height due to dilution despite the production of sulfuric acid. In turn, increased pH promotes oxidation by $O_3$ in the upper parts of the cloud, whereas oxidation by $H_2O_2$ dominates in lower parts of the cloud, according to their study.

Case1 shown in Fig. 5b represents a hypothetical "no oxidation" scenario where all physical and chemical conditions are the

same as in base case, the reversible chemical processes are allowed, and oxidation is prohibited. The scenario without oxidation is overall less acidic than base case (Fig. 5a). Additionally, without oxidation there is no difference between the pH values in the updraft and downdraft in Fig. 5b. Without oxidation, all the chemical processes are reversible and the dissolved chemical compounds are outgassed to the atmosphere as the cloud droplets evaporate in the downdraft.

Case2 differs from base case by increasing the initial $NH_3$ volume fraction from 0.1 ppb-v to 0.4 ppb-v (see Tab. 2 and

Tab. 3). Because the initial aerosol particle size distribution is the same as in base case, the mean aerosol and droplet sizes and concentrations at the end of the simulation are not different from base case (not shown). Figure 5c shows the liquid water weighted average pH for case2. The average pH in case2 (Fig. 5c) is higher than in base case (Fig. 5a), that is, both cloud droplets and rain drops are less acidic in case2 than in base case. In contrast to base case, in the updraft (left-hand side of the plots), the pH in case2 actually decreases with height above the cloud base. This is because the higher initial $NH_3$ volume

fraction increases its uptake and counters the low pH values caused by initial acidic aerosol particles. Then, as the water drops are advected upwards, oxidation produces sulfuric acid and the average pH decreases. Near the cloud top, the $NH_3$ is degassed back to the environment. Case2 results are in agreement with the trajectory ensemble model simulations by Zhang et al. (1999). In their study, the initial aerosol size distribution is the same as in base case and case2. However, their initial trace gas volume fractions are much higher and aim to represent a "moderately polluted marine environment" (their base case $NH_3$

volume fraction is ten times larger than base case value assumed here). As in case2 presented here, the high initial $NH_3$ volume fractions in Zhang et al. (1999) increase the pH near the cloud base and promote oxidation by $O_3$ during the first minutes after the simulated parcels entered the cloud. Because the sulfuric acid was produced, the pH dropped and oxidation by $H_2O_2$ becomes dominant in the higher regions of the cloud, as reported in their study.

Case3 increases the initial aerosol concentration to $150 \, cm^{-3}$, while keeping all other initial conditions the same as in base

case (see Tab. 2 and Tab. 3). In general, higher initial aerosol particle concentration results in higher cloud droplet concentrations. This in turn creates smaller cloud droplet effective radii that virtually prohibits the onset of precipitation during the 30 minutes simulation time (not shown). Figure 5d shows the liquid water weighted average pH for case3. Similar to base case (Fig. 5a), the pH increases with height due to the dilution and the downdraft droplets are more acidic due to the ongoing oxidation. However, case3 is more acidic than base case because the overall droplet sizes are smaller and they are therefore



less diluted.

At the end of base case simulation, 18% of the total available $S^{IV}$ is oxidized. As a result, 0.14 $\mu$g/m$^3$ of dry particulate matter are created during oxidation (an average value for the whole computational domain reported in relation to the dry air volume). In total, 40% of the final dry particulate matter is created due to oxidation and 60% originates from the initial aerosol mass. The oxidation is a significant source of dry particulate matter because the initial aerosol mass is very low (only 0.21 $\mu$g/m$^3$ dry air). Oxidation by $H_2O_2$ is the dominant path: 92% of the $S^{VI}$ molecules created during oxidation are oxidized by $H_2O_2$. More alkaline conditions of case2 enhance the efficiency of oxidation. At the end of case2, 21% of available $S^{IV}$ is oxidized. As a result, oxidation produces 0.16 $\mu$g of dry particulate matter per m$^3$ of dry air (average over the whole computational domain). For case2, 44% of the final dry particulate matter is created due to oxidation and 56% originates from the initial ammonium bisulfate aerosol. Similarly to base case, the significance of oxidation as a source of dry particulate matter is caused by a very low initial aerosol mass. Due to more alkaline conditions, oxidation by $O_3$ becomes more important than in base case. At the end of case3 simulation, 39% of the $S^{VI}$ molecules that are created during oxidation are produced by the $O_3$ path and 61% by the $H_2O_2$ path. In contrast, more acidic conditions of case3 hinder the $O_3$ reaction path. Virtually all molecules of sulfate that are created during oxidation are oxidized by $H_2O_2$. As a result, the conversion of sulfur to sulfate is slightly less effective in case3. At the end of case3 simulation, 17% of available $S^{IV}$ is oxidized. As a result 0.13 $\mu$g of dry particulate matter are created per m$^3$ of dry air. At the end of case3 simulation, 17% of the dry particulate matter is created by oxidation and 83% originates from the initial aerosol. The initial aerosol mass is larger in case3 than in base case due to the higher initial aerosol concentration (case3 contains initially 0.61 $\mu$g/m$^3$ of dry particulate matter). Because of this, even though the produced sulfate mass is only slightly lower than in base case, the relative importance of oxidation decreases by more than 20 percentage points in case3. Due to the simple kinematic setup chosen in this study the values reported here cannot be treated as representative for the atmospheric conditions. They are shown to allow comparison between base case and the sensitivity test cases.

Finally, the impact of collisions and aqueous phase oxidation of sulfur on the aerosol and water drop size distributions is examined. For this purpose, the aerosol particle size distributions from base case (Fig. 6a) and case3 (Fig. 6b) are compared. The black line represents the initial aerosol size distribution and the green and red lines represent the final aerosol size distribution for in-cloud ($r_c > 0.01$ g/kg) and precipitating ($r_r > 0.01$ g/kg) grid-cells. The two cases are chosen because they have different initial aerosol size distributions. In both cases the cloud-processed aerosol size distributions (green and red lines) have a bi-modal shape. This is a footprint of oxidation that creates the Hoppel minimum in the dry radius size distribution. The same effect is obtained in the adiabatic parcel tests discussed in Sec. 4 and shown in Fig. 3. Moreover, the efficient collisions between water drops in base case create a tail of bigger aerosol sizes in Fig. 6a. The effect is stronger for the precipitating grid-cells (red line). In case3 fewer collisions between water drops occur than in base case and therefore no precipitation and no tail of big aerosol particles is created. Also, in case3, the change in size distribution of aerosol particles caused by oxidation is smaller because the produced sulfate is divided among bigger number of aerosol particles.





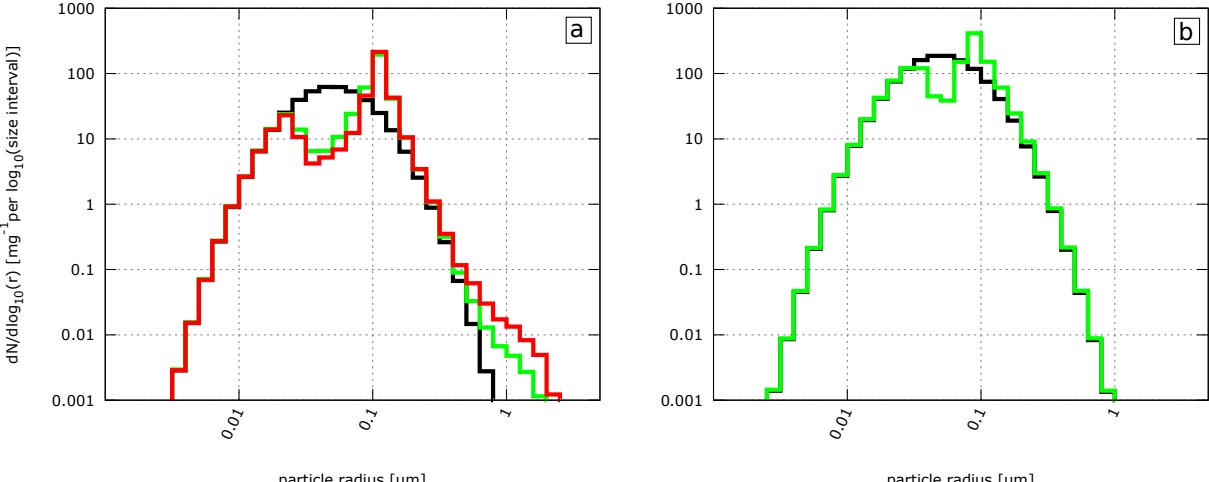

**Figure 6.** Size distributions of dry radii for base case (a) and case3 (b). The initial dry radius size distribution is marked in black, final dry radius size distribution from grid-cells with $r_c > 0.01$ g/kg in green and from grid-cells with $r_r > 0.01$ g/kg in red. See Tab. 2 and Tab. 3 for a definition of simulation setups.

## 6  Summary and outlook

The work presented here describes a new extension of the *libcloudph++* that allows including aqueous phase chemical reactions within water drops in the Lagrangian microphysics scheme. The extension covers the aqueous phase oxidation of sulfur to sulfate. The modular way in which the library is implemented along with the provided documentation should allow, if needed,

further development to cover more chemical compounds and reactions. The 0-dimensional and 2-dimensional tests described in this work as well as comparison with other numerical studies using bin microphysics schemes along with aqueous chemistry representation document the correctness of the design and the implementation of the LMC scheme. Additionally, the changes in the user interface due to aqueous chemistry extension are described in Sec. A in the Appendix. Section C in the Appendix completes the description with a list of chemical constants used in the library and chemical reactions included.

The models used in this study to test the chemistry scheme provide a simplified view of the macrophysical cloud properties. They enable validation and testing of the Lagrangian scheme but do not provide a good balance between the representation of cloud microphysics and dynamics. As a next step, the Lagrangian scheme needs to be coupled to an eddy-resolving model. This would allow quantifying how microphysical and chemical processes affect precipitation in the model and how they affect the cloud lifetimes simulated by the model.

*Code availability.* The *libcloudph++* library along with the aqueous-phase chemistry extension, the parcel model and the 2D slice model are released under GNU General Public License v3.0. The version of *libcloudph++* accompanying this publication is tagged as "1.1.0" at



```
template<typename real_t>
struct opts_t
{
  // process toggling for chemistry
  bool chem_dsl, chem_dsc, chem_rct;

  // ...
```

**Figure A1. lgrngn::opts_t definition**

the project repository and is also available as an electronic supplement to this paper. *libcloudph++* and the 2D slice model are available at: https://github.com/igfuw/libcloudphxx and the parcel model is available at: https://github.com/igfuw/parcel.

The supported platforms are: Linux with GNU g++11, Linux with LLVM clang++12 and Apple OSX with the Apple clang++13 (tested using continuous integration framework).

## 5  Appendix A:  User Interface

The user interface of the Lagrangian microphysics scheme of *libcloudph++* is presented in Sec 5.2. in Arabas et al. (2015). Here, additional information related to the new aqueous phase chemistry scheme is provided. The *libcloudph++* is implemented in C++ and therefore some nomenclature related to this programming language is used. For a thorough introduction to C++ programming language see Stroustrup (2013).

The aqueous chemistry module is implemented as an optional extension to the Lagrangian microphysics scheme in *libcloudph++*. It uses the same **libcloudphxx::lgrngn** namespace as the original scheme. Again the template parameter **real_t** selects between floating point formats of simulations. The Lagrangian microphysics scheme options are grouped into a structure named **lgrngn::opts_t**. Chemistry module adds three Boolean fields to this structure: **chem_dsl chem_dsc** and **chem_rct**, see code listing in Fig. A1. When set to true by the user, they switch on dissolving of trace gases into water drops, dissociation of chemical compounds in water drops and oxidation reaction, respectively. The parameters in **lgrngn::opts_t** can be changed during simulation. For example during the 2-dimensional kinematic simulations from Sec. 5, oxidation is enabled by setting the **chem_rct** parameter to true at the end of spin-up. Other parameters that cannot be changed during simulation are encapsulated in **lgrngn::opts_init_t** structure. Chemistry module adds three fields to this structure: (i) A Boolean **chem_switch** field that enables memory allocation for additional variables needed for chemistry representation. (ii) An integer **sstp_chem** field that defines the number of sub-steps to be carried out in aqueous chemistry calculations. (iii) A **real_t chem_rho** field that defines the dry aerosol density, see code listing in Fig. A2.

The names of chemical compounds available in the aqueous phase chemistry module are stored in a **chem_species_t** enumerator, see code listing in Fig. A3. The state of all variables used by the Lagrangian scheme is stored in an instance of the **lgrngn::particles_t** structure shown in code listing in Fig. A4. The second template parameter of that structure selects between





```cpp
template<typename real_t>
struct opts_init_t
{
  // if false no chemical reactions are allowed
  // (no memory allocation)
  bool chem_switch;
  // substeps for chemistry
  int sstp_chem;
  // assumed dry aerosol density
  real_t chem_rho;
}
```

**Figure A2. lgrngn::opts_init_t definition**

```cpp
enum chem_species_t
{
  // both gas and total dissolved chem species
  HNO3, NH3, CO2, SO2, H2O2, O3,
  // created sulfate
  S_VI,
  // additional H+ for pH
  H
}
```

**Figure A3. lgrngn::chem_species_t definition**

CPU and GPU calculations (see the discussion in Sec. 5.2 in Arabas et al., 2015, for details). The initialization, time-stepping and output from the Lagrangian scheme are done using the methods of **lgrngn::particles_t** structure. Their signatures are provided in code listing in Fig. A4.

The **init()** method performs initialization and should be called first. As discussed in Arabas et al. (2015) the first three arguments are obligatory and should point to the dry air potential temperature, water vapor mixing ratio and dry air density fields of the driver model that uses the *libcloudph++*. The next three arguments should point to the Courant number field components. They are optional and depend on the dimensionality of the solved problem. For example, for the parcel model tests from Sec. 4 none are necessary, whereas for the 2-dimensional kinematic model from Sec. 5 two arguments are specified in order to de-
scribe the velocity field. The last argument of **init()** is a map with keys from **chem_species_t** enumerator and values pointing to the corresponding trace gas mixing ratio fields from the driver model. This is an optional argument for simulations with aqueous phase chemistry.

During time-stepping the Lagrangian scheme computations are performed by **step_sync()** and **step_async()** methods. The
first one gathers all the processes that affect the driver model fields (such as condensation/evaporation or aqueous phase chem-



```
template <typename real_t, backend_t backend>
struct particles_t: particles_proto_t<real_t>
{
  // initialisation
  void init(
    const arrinfo_t<real_t> th,
    const arrinfo_t<real_t> rv,
    const arrinfo_t<real_t> rhod,
    const arrinfo_t<real_t> courant_x,
    const arrinfo_t<real_t> courant_y,
    const arrinfo_t<real_t> courant_z,
    const std::map<
      enum chem_species_t,
      const arrinfo_t<real_t>
    > ambient_chem
  );

  // time-stepping methods
  void step_sync(
    const opts_t<real_t> &,
    arrinfo_t<real_t> th,
    arrinfo_t<real_t> rv,
    const arrinfo_t<real_t> rhod,
    const arrinfo_t<real_t> courant_x,
    const arrinfo_t<real_t> courant_y,
    const arrinfo_t<real_t> courant_z,
    std::map<
      enum chem_species_t,
      arrinfo_t<real_t>
    > ambient_chem,
  );

  void step_async(
    const opts_t<real_t> &
  );

  // diagnostic methods
  // ...
  void diag_chem(const enum chem_species_t&);
  // ...
```

Figure A4. lgrngn::particles_t definition





istry) and the second one gathers all the processes that can be calculated asynchronously (for example collisions or sedimentation). The list of arguments of **step_sync()** method is extended by the chemistry module. Similar to the **init()** method, a map linking **chem_species_t** enumerator items with the driver model mixing ratio fields needs to be provided as the last optional argument. The Lagrangian scheme overwrites the driver model fields during simulation. The signature of **step_async** method

5   is not changed by the new chemistry module.

As discussed in Arabas et al. (2015), the **lgrngn::particles_t** structure provides many methods for obtaining statistical information on the SD parameters (prefixed with **diag**). The chemistry model adds to them the **diag_chem** method that outputs the total mass of a chemical compound dissolved into droplets. The chemical compound is selected using the **chem_species_t**

10  enumerator items. See the discussion in Sec. 5.2 in Arabas et al. (2015) for the details on how to select the size ranges of droplets specified for output or how to output other statistical parameters.





## Appendix B:  Glossary

|  |  |
|---|---|
| $\alpha_M$ | mass accommodation coefficient of water vapor |
| $\alpha_{M_A}$ | mass accommodation coefficient of the chemical compound "A" |
| $c_{A\infty}$ | ambient concentration of the trace gas "A" |
| $D_A$ | diffusion coefficient of the chemical compound "A" |
| $E$ | reaction activation energy |
| $\mathbb{H}_A^{eff}$ | effective equilibrium constant for dissolution of the chemical compound "A" |
| $\Delta H_D$ | reaction enthalpy of dissociation at constant temperature and pressure |
| $\Delta H_H$ | reaction enthalpy of dissolution at constant temperature and pressure |
| $\mathbb{K}_A$ | dissociation constant |
| $k_{0,...,4}$ | reaction rate coefficients |
| $\kappa$ | hygroscopicity parameter |
| $M_A$ | molar mass of the chemical compound "A" |
| $n(r_d)$ | spectral density function of aerosol particle sizes |
| $n_{tot}$ | total aerosol concentration |
| $\mathcal{N}$ | super-droplet multiplicity |
| $\theta$ | dry air potential temperature |
| $\mathbb{R}_A$ | reaction rate of the chemical compound "A" |
| $\rho_d$ | dry air density |
| $r_d$ | dry radius |
| $\overline{r_d}$ | mean radius of the assumed lognormal aerosol particle size distribution |
| $r_w$ | drop radius |
| $r_c$ | cloud water mixing ratio |
| $r_r$ | rain water mixing ratio |
| $r_v$ | water vapor mixing ratio |
| $\sigma_g$ | geometric standard deviation |
| $<v>$ | average velocity of the molecules |

## Appendix C:  List of chemical compounds and constants





**Table C1.** Chemical compounds considered in this work.

| chemical compound | formula | molar mass (g moles$^{-1}$) | source |
|---|---|---|---|
| ammonia | $NH_3$ | 17 | trace gas |
| carbon dioxide | $CO_2$ | 44 | trace gas |
| hydrogen peroxide | $H_2O_2$ | 34 | trace gas |
| nitric acid | $HNO_3$ | 63 | trace gas |
| ozone | $O_3$ | 48 | trace gas |
| sulfur dioxide | $SO_2$ | 64 | trace gas |
| sulfuric acid | $H_2SO_4$ | 98 | oxidation reaction product |
| ammonium bisulfate | $NH_4HSO_4$ | 115 | initial aerosol |

**Table C2.** Dissociation constants and their temperature correction coefficients (taken from Kreidenweis et al., 2003). Dissociation of $H_2SO_4$ is taken from Tab. 6.A.1 in Seinfeld and Pandis (1998).

| | equilibrium reaction | dissociation constant at 298K (moles liter$^{-1}$) | temp. corr. $\frac{-\Delta H_D}{R}$ (K) |
|---|---|---|---|
| $\mathbb{K}_{HNO3}$ | $HNO_3(aq) <--> H^+ + NO_3^-$ | 15.4 | 8700 |
| $\mathbb{K}_{SO2}$ | $SO_2 * H_2O <--> H^+ + HSO_3^-$ | $1.3 \times 10^{-2}$ | 1960 |
| $\mathbb{K}_{NH3}$ | $NH_3 * H_2O <--> NH_4^+ + OH^-$ | $1.7 \times 10^{-5}$ | $-450$ |
| $\mathbb{K}_{CO2}$ | $CO_2 * H_2O <--> H^+ + HCO_3^-$ | $4.3 \times 10^{-7}$ | $-1000$ |
| $\mathbb{K}_{HSO3}$ | $HSO_3^- <--> H^+ + SO_3^{2-}$ | $6.6 \times 10^{-8}$ | 1500 |
| $\mathbb{K}_{HCO3}$ | $HCO_3^- <--> H^+ + CO_3^{2-}$ | $4.68 \times 10^{-11}$ | $-1760$ |
| | $H_2SO_4(aq) <--> H^+ + HSO_4^-$ | $\infty$ | - |
| $\mathbb{K}_{H2SO4}$ | $HSO_4^- <--> H^+ + SO_4^{2-}$ | $1.2 \times 10^{-2}$ | 2720 |



**Table C3.** Dissolution constants and their temperature correction coefficients (taken from Kreidenweis et al., 2003).

|  | equilibrium reaction | dissolution constant at 298K (moles liter$^{-1}$ atm$^{-1}$) | temp. corr. $\frac{-\Delta H_H}{R}$ (K) |
|---|---|---|---|
| $\mathbb{H}_{HNO3}$ | $HNO_3(g) <--> HNO_3(aq)$ | $2.10 \times 10^5$ | - |
| $\mathbb{H}_{H2O2}$ | $H_2O_2(g) <--> H_2O_2(aq)$ | $7.45 \times 10^4$ | 7300 |
| $\mathbb{H}_{NH3}$ | $NH_3(g) <--> NH_3 * H_2O$ | 62 | 4110 |
| $\mathbb{H}_{SO2}$ | $SO_2(g) <--> SO_2 * H_2O$ | 1.23 | 3150 |
| $\mathbb{H}_{CO2}$ | $CO_2(g) <--> CO_2 * H_2O$ | $3.40 \times 10^{-2}$ | 2440 |
| $\mathbb{H}_{O3}$ | $O_3(g) <--> O_3(aq)$ | $1.13 \times 10^{-2}$ | 2540 |

**Table C4.** Diffusion constants (Massman, 1998; Tang et al., 2014) and accommodation coefficients (Kreidenweis et al., 2003) for relevant chemical compounds .

|  | diffusion coeff. $D_A$ (m$^2$/s) | mass accommodation coeff. $\alpha_{M_A}$ |
|---|---|---|
| $HNO_3$ | $65.25 \times 10^{-6}$ | 0.05 |
| $H_2O_2$ | $87.00 \times 10^{-6}$ | 0.018 |
| $NH_3$ | $19.78 \times 10^{-6}$ | 0.05 |
| $SO_2$ | $10.89 \times 10^{-6}$ | 0.035 |
| $CO_2$ | $13.81 \times 10^{-6}$ | 0.05 |
| $O_3$ | $14.44 \times 10^{-6}$ | 0.00053 |

**Table C5.** Reaction rate coefficients and their temperature correction coefficients (taken from Kreidenweis et al., 2003).

| oxidation reaction path | reaction rate coefficient (liter moles$^{-1}$ s$^{-1}$) at 298K | temperature correction $\frac{-E}{R}$ (K) |
|---|---|---|
| $O_3(aq) + SO_2 * H_2O -> S^{VI}$ | $k_0 = 2.4 \times 10^4$ | 0 |
| $O_3(aq) + HSO_3^- -> S^{VI}$ | $k_1 = 3.5 \times 10^5$ | -5530 |
| $O_3(aq) + SO_3^{2-} -> S^{VI}$ | $k_2 = 1.5 \times 10^9$ | -5280 |
| $H_2O_2(aq) + HSO_3^- -> S^{VI}$ | $k_3 = 7.45 \times 10^7$ | -4430 |



*Competing interests.* The authors declare that they have no competing interests

*Acknowledgements.* The work was funded by Poland's National Science Centre (Narodowe Centrum Nauki), decisions 2012/06/M/ST10/00434
and 2014/15/N/ST10/05143. We would also like to thank Travis CI and GitHub for providing their platforms free of charge for academic
purposes and open source projects respectively.



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
