# Peer review of "libcloudph++ 2.0: aqueous phase chemistry extension of the particle-based cloud microphysics scheme"

_Geoscientific Model Development, 2018_

## Referee Comment (RC1) · Anonymous Referee #1 · 20 May 2018

**Review of "libcloudph++ 1.1: aqueous phase chemistry extension of the Lagrangian cloud microphysics scheme" by Jaruga and Pawlowska (GMD-2018-96)**

The manuscript describes and validates the addition of aqueous sulfur chemistry to the *libcloudph++* Lagrangian cloud model (LCM) (Arabas et al., 2015), which are a new class of recently developed highly detailed microphysical models. The presented test cases are an adiabatic parcel simulation based upon the intercomparison by Kreidenweis et al. (2003) and a kinematic two-dimensional stratocumulus study. This manuscript presents a profound description of consideration of aqueous sulfur chemistry in the *libcloudph++* LCM, which will allow much more sophisticated investigations than with earlier models. (However, the manuscript misses the opportunity to demonstrate all possible advantages of this great new development.) All in all, the manuscript fits the scope of Geoscientific Model Development and I recommend its publication once my comments are addressed.

**A Slightly Major Comment**

P. 8, l. 2 - p. 9, l. 7, Fig. 1: The liquid water content is one of the most important parameters to steer aqueous sulfur chemistry (and probably in cloud physics in general). Therefore, it is necessary that the same liquid water content is reproduced by the presented model to validate it with the intercomparison results of Kreidenweis et al. (2003), in which seven other models are capable of reproducing the same liquid water content. The authors state that the reason for this discrepancy results from the unspecified pressure profile in the study of Kreidenweis et al. (2003). However, there might be another potential source for the deviation: it could be the slightly too high initial temperature in the current study, which is stated as 285.5 K (Tab. 1 of the manuscript) but 285.2 K in Kreidenweis et al. (2003, their Tab. 3). Moreover, it might be helpful to clarify the calculation of the pressure (and density) profile by using equations (the description on p. 8, ll. 21-29 is hard to follow). Finally, why does the axis stating the height above cloud base start with 100 m but the time above cloud base with 0 s (Fig. 1)?

**Minor Comments**

Introduction: Many previous models (so-called "moving bin" or "moving grid" models) used a Lagrangian framework for the representation of aqueous sulfur chemistry as done in this study. Accordingly, they are based on similar (sometimes identical) model equations as the described in the current manuscript. As the LCM, these "moving bin" models do not suffer from numerical diffusion (e.g., p. 4, ll. 7-8). However, these models are only applicable in simplified parcel or trajectory simulations (e.g., Feingold et al., 1998; Kreidenweis et al., 2003). The authors should make clear that their main advancement is the possibility to use a Lagrangian representation of aqueous sulfur chemistry in multidimensional simulations with a fully coupled dynamics model, which hasn't been possible before.

P. 2, ll. 8-24: Please rearrange/rewrite these two paragraphs. Especially the first paragraph gives no real reasons why "aqueous phase oxidation of sulfur" is an important chemical reaction, it only states it is a very common reaction. Moreover, there are many interesting points in these two paragraphs but it would be nice to connect them more closely with the presented study/model.

P. 2, ll. 28–29: What about the collection of submicron aerosol by droplets (e.g., Ardon-Dryer et al. (2015), Ladino et al. (2011))? These processes are not considered in the current model but also affect the size of aerosols.

P. 3, l. 2: What is meant by an "atmospheric aerosol particle" in this context? There must certainly be a size restriction to the cited statement since very small aerosols are not cloud-active.

P. 3, ll. 5-6: The effect of collision and coalescence on the amount of aerosol inside a cloud droplet (in absence of aqueous sulfur chemistry!) has been considered in an LCM framework before (see Hoffmann, 2017).

P. 4, ll. 11-12: This is a vague statement since it depends on the equations considered and the applied numerical solvers.

P. 5, l. 2: How does the wind speed affect the chemical scheme?

P. 5, ll. 8-13: Does this restriction affect the results? And if yes, how? And what happens to giant and ultra-giant aerosols, which remain less diluted throughout their lifecycle?

P. 5, ll. 20-22: In what aspect is the LCM beneficial? Using a low number of super-droplets, the LCM will suffer from the same problems as a "moving bin" scheme.

P. 5, Eq. (1): For a reaction $ab \rightleftharpoons a + b$ dissociation constant is defined as $\mathbb{K}_{ab} = [a][b]/[ab]$. Accordingly, Eq. (1) is just a relationship to adjust a measured dissociation constant $\mathbb{K}_{ab}$ to a given temperature. Denote the meaning of $T_0$ and $R$.

P. 6, Eq. (2): I understand that this equation is based on the neutral charge condition. For clarity, it might be necessary to comment on how the expressions for the individual terms are derived. (Or state an appropriate reference.)

P. 6, ll. 15-16; p. 7, l. 15: How is this "small" timestep determined? Is there a criterion? How does the user know that an appropriate timestep is chosen?

P. 7, ll. 26-28: Why is $H_2SO_4$ used instead of $S^{IV}$ to denote sulfuric acid? Please clarify.

P. 8, ll. 4-5: What is meant by "initial conditions above supersaturation"?

P. 8, ll. 15-20; p. 12, Fig. 3: How is the aerosol distribution transformed to superdroplets? Is the distribution binned and a superdroplet represents each bin or is a random generator used that follows the given distribution? This is in important information to understand the slight fluctuations in the initial spectrum (Fig. 3).

P. 10, ll. 9-10: How is it possible to have activated particles at a relative humidity of 100 %? To activate a droplet, a *super*saturation (RH>1) is necessary.

P. 13, ll. 3-4: The Hall (1980) collection efficiencies are not very suitable for big droplets since they neglect the effect of a coalescence efficiency of less than unity. How are "big" droplets defined?

P. 16, Fig. 4b: What is meant by "spec." in the panel's title?

P. 18, l. 5: Eq. (5) shows that the oxidation of $H_2O_2$ is not independent of the acidity ($[H^+]$ in the denominator). (Although the dependence might be very weak.)

P. 18, l. 19-20: "This is because … aerosol particles." Is this a direct result of Eq. (2)?

**Technical Comments**

P. 1, ll. 20-21: "clearly defined user interface" is a slightly subjective statement.

P. 1, l. 21: Make clear that "separation of concerns" is a software design principle.

P. 2, l. 17: Define IPCC.

P. 2, ll. 33-34: All these references should be in one pair of parentheses.

P. 3, l. 20-22: As a service to the reader, it would be nice to state how many new attributes are added for the consideration of aqueous sulfur chemistry.

P. 4, l. 15: The citation is "Lee et al. (2014)" not "Junghwa et al. (2014)".

P. 4, ll. 22-23: The pH is a scale used to quantify the acidity of a solution. It is (linguistically) awkward to keep track of a scale and not the underlying physical meaning.

P. 6, Eq. (2): The underbrace needs to include all terms on the right-hand side of the equation.

P. 10, l. 4: These values are stated in Tab. 2 of Kreidenweis et al. (2003).

P. 12, ll. 7-8, 15: Semicolons in an enumeration of references are unusual.

P. 13, l. 21: "the the"

P. 15, l. 27, 29: Is Fig. 4f addressed of Fig. 4c?

**References**

Arabas, S., Jaruga, A., Pawlowska, H., and Grabowski, W. W. (2015). libcloudph++ 1.0: a single-moment bulk, double-moment bulk, and particle-based warm-rain microphysics library in C++. Geoscientific Model Development, 8(6), 1677.

Ardon-Dryer, K., Huang, Y. W., and Cziczo, D. J. (2015). Laboratory studies of collection efficiency of sub-micrometer aerosol particles by cloud droplets on a single-droplet basis. Atmospheric Chemistry and Physics, 15(16), 9159-9171.

Feingold, G., Kreidenweis, S. M., and Zhang, Y. (1998). Stratocumulus processing of gases and cloud condensation nuclei: 1. Trajectory ensemble model. Journal of Geophysical Research: Atmospheres, 103(D16), 19527-19542.

Hall, W. D. (1980). A detailed microphysical model within a two-dimensional dynamic framework: Model description and preliminary results. Journal of the Atmospheric Sciences, 37(11), 2486-2507.

Hoffmann, F. (2017). On the limits of Köhler activation theory: how do collision and coalescence affect the activation of aerosols?. Atmospheric Chemistry and Physics, 17(13), 8343-8356.

Kreidenweis, S.M., Walcek, C.J., Feingold, G., Gong, W., Jacobson, M.Z., Kim, C.H., Liu, X., Penner, J.E., Nenes, A., and Seinfeld, J.H. (2003). Modification of aerosol mass and size distribution due to aqueous-phase SO2 oxidation in clouds: Comparisons of several models. Journal of Geophysical Research: Atmospheres, 108(D7).

Ladino, L., Stetzer, O., Hattendorf, B., Günther, D., Croft, B., and Lohmann, U. (2011). Experimental study of collection efficiencies between submicron aerosols and cloud droplets. Journal of the Atmospheric Sciences, 68(9), 1853-1864.

Lee, J., Noh, Y., Raasch, S., Riechelmann, T., and Wang, L. P. (2014). Investigation of droplet dynamics in a convective cloud using a Lagrangian cloud model. Meteorology and Atmospheric Physics, 124(1-2), 1-21.

---

## Author Comment (AC1) · 24 Jul 2018

The reply to both reviews is submitted as a supplement. The reply includes the revised manuscript with highlighted changes.

Please also note the supplement to this comment:
https://www.geosci-model-dev-discuss.net/gmd-2018-96/gmd-2018-96-AC1-supplement.pdf

---

## Author Response (AR1)

We would like to thank the two anonymous Referees for their comments on the manuscript. In addition, we would like to thank Dr Sylwester Arabas for his review and suggestions. All comments have been addressed in the revised version of the manuscript. Below we are submitting our replies to the Reviewers, the list of additional changes and the revised version of the manuscript.

**Review of libcloudph++ 1.1: aqueous phase chemistry extension of the Lagrangian cloud microphysics scheme by Jaruga and Pawlowska (GMD-2018-96)', Anonymous Referee #1**

- **P. 8, l. 2 - p. 9, l. 7, Fig. 1: The liquid water content is one of the most important parameters to steer aqueous sulfur chemistry (and probably in cloud physics in general). Therefore, it is necessary that the same liquid water content is reproduced by the presented model to validate it with the intercomparison results of Kreidenweis et al. (2003), in which seven other models are capable of reproducing the same liquid water content. The authors state that the reason for this discrepancy results from the unspecified pressure profile in the study of Kreidenweis et al. (2003). However, there might be another potential source for the deviation: it could be the slightly too high initial temperature in the current study, which is stated as 285.5 K (Tab. 1 of the manuscript) but 285.2 K in Kreidenweis et al. (2003, their Tab. 3).**

  Thank you for catching this error. This is just a typo in the manuscript and the actual initial temperature used in the parcel model simulations was 285.2K: see `https://github.com/igfuw/parcel/blob/master/chem_conditions.py#L13`.

  We have tested the dependence of the parcel model on different pressure $p$ and dry air density $\rho_d$ calculations. We repeated the parcel model simulations using three different ways to calculate $p$ and $\rho_d$. In all simulations $p$ is calculated by integrating the hydrostatic equation:

  $$\frac{dp}{dz} = -\rho g, \tag{1}$$

  where $p$ represents pressure, $z$ is the vertical displacement, $\rho$ is the density of air and $g$ is the gravitational acceleration.

  - In the first simulation, to integrate Eq. (1) we assume that $\rho$ is constant and equal $\rho_0 = 1.15$ kg m$^{-3}$. The assumed density value is taken from Tab 3 in Kreidenweis et al. (2003). At each time level $n$ the pressure is given by

    $$p^n = p^0 - \rho_0 g z^n. \tag{2}$$

  The dry air density is given by:

  $$\rho_d(p, \theta, r_v) = \frac{p - p_v(p, r_v)}{R_d \theta (\frac{p}{p_{1000}})^{\frac{R_d}{c_{pd}}}}, \tag{3}$$

  where $p_v(p, r_v)$ represents partial pressure of water vapor, $p_{1000}$ stands for pressure equal 1000 $hPa$ that comes from the definition of potential temperature, $R_d$ is the gas constant for dry air and $c_{pd}$ is the specific heat at constant pressure for dry air. The dry air density at a given time level $n$ is calculated as $\rho_d(p^n, \theta^n, r_v^n)$. This simulation is labeled **const_rho**

  - For the second simulation, we assume that the density is constant at each height level (i.e. piecewise constant profile). As a result, at a given time level $n$ the pressure used to predict $\rho_d$ is defined as

    $$p^n = p^{n-1} - \rho^{n-1} g z^n. \tag{4}$$

  Again the dry air density at a given time level $n$ is calculated using Eq.(3) as $\rho_d(p^n, \theta^n, r_v^n)$. This simulation is labeled **piecewise_const_rho** and was used in the submitted manuscript.

  - In the third simulation, we assume that the potential temperature, $\theta = \theta^0$, and water vapor mixing ratio, $r_v = r_v^0$ are constant. At a given time level $n$ $p$ is then defined as

    $$p^n = p_{1000} \left( (\frac{p^0}{p_{1000}})^{\frac{R_d}{c_{pd}}} - \frac{R_d g(z^n - z^0)}{c_{pd} \theta^0 R(r_v^0)} \right)^{\frac{c_{pd}}{R_d}}, \tag{5}$$

  where $R(r_v^0)$ is the gas constant for moist air. The dry air density at a given time level $n$ is calculated using Eq.(3) as $\rho_d(p^n, \theta^0, r_v^0)$. This method follows the procedure used in the kinematic model, described

in Arabas et al. (2015) and used in the manuscript in the 2-dimensional simulations. This simulation is labeled **const_th_rv**

The results of the tests are presented in Fig. 1 and can be compared with Fig. 1 in Kreidenweis et al. (2003). Additionally the values of parcel acidity, sulfate formation, number of droplets at maximum saturation level and the maximum saturation are reported in Tab. 1. The simulation results are similar for the $SO_2$ concentration and pH calculations. The const_th_rv simulation is the biggest outlier. The biggest differences can be seen for the liquid water content profile (first panel) of Fig. 1. In general, the const_rho simulation agrees best with the results shown in Fig. 1 in Kreidenweis et al. (2003). In order to better match the results presented in Kreidenweis et al. (2003) the parcel model simulations shown in the manuscript are changed to use the constant density assumption (const_rho). The relevant descriptions and figures are changed.

[Figure]

Figure 1: The parcel model simulations using three different ways to calculate the pressure and dry air density profiles: assuming constant density profile (blue), assuming piecewise constant density profile (red) and assuming constant potential temperature and water vapor mixing ratio profile (green). The panels show the liquid water mixing ratio, the $SO_2$ concentration and the water volume weighted average pH, respectively.

Table 1: The microphysical and chemical characteristics of the parcal model simulations.

|  | const_rho | piecewise_const_rho | const_th_rv | Kreidenweis et al. (2003) |
|---|---|---|---|---|
| water volume weighted pH | 4.86 | 4.83 | 4.79 | 4.82 - 4.85 |
| sulfate formation (ppt) | 171 | 168 | 161 | 170 - 180 |
| sulfate formation by $H_2O_2$ (ppt) | 99 | 100 | 104 | 85 - 105 |
| sulfate formation by $O_3$ (ppt) | 72 | 68 | 57 | 70 - 85 |
| number of droplets in $cm^{-3}$ | 269 | 272 | 265 | 275 - 358 |
| maximum supersaturation | 0.27 | 0.27 | 0.26 | 0.23 - 0.37 |

**Moreover, it might be helpful to clarify the calculation of the pressure (and density) profile by using equations (the description on p. 8, ll. 21-29 is hard to follow).**

The corresponding description was changed to reflect the changed way of calculating pressure and density profile.

**Finally, why does the axis stating the height above cloud base start with 100 m but the time above cloud base with 0 s (Fig. 1)?**

Thank you, we have corrected the axis labels.

- **Introduction: Many previous models (so-called moving bin or moving grid models) used a Lagrangian framework for the representation of aqueous sulfur chemistry as done in this study. Accordingly, they are based on similar (sometimes identical) model equations as the described**

**in the current manuscript. As the LCM, these moving bin models do not suffer from numerical diffusion (e.g., p. 4, ll. 7-8). However, these models are only applicable in simplified parcel or trajectory simulations (e.g., Feingold et al., 1998; Kreidenweis et al., 2003). The authors should make clear that their main advancement is the possibility to use a Lagrangian representation of aqueous sulfur chemistry in multidimensional simulations with a fully coupled dynamics model, which hasnt been possible before.**

Thank you for pointing this out! Using the term "Lagrangian" can be indeed confusing as it might refer to different microphysics schemes: (i) "Lagrangian-in-droplet-radius" - as in the "moving bin" models from Kreidenweis et al. (2003) or (ii) "Lagrangian-in-droplet-radius-and-space" - as it is done in the super-droplet method (Shima et al., 2009) used in this study. To avoid confusion we added a clarification in the introduction. We also decided to stop using the term Lagrangian when referring to our scheme. Instead, following the notation used in the original super-droplet paper by Shima et al. (2009), we will refer to this method as particle-based.

- **P. 2, ll. 8-24: Please rearrange/rewrite these two paragraphs. Especially the first paragraph gives no real reasons why aqueous phase oxidation of sulfur is an important chemical reaction, it only states it is a very common reaction. Moreover, there are many interesting points in these two paragraphs but it would be nice to connect them more closely with the presented study/model.**

  We rearranged the paragraphs to hopefully put more emphasis on the importance of the aqueous phase oxidation of sulfur. Overall, it is worth underlining that the main reason for adding the aqueous phase oxidation of sulfur to the particle-based microphysics scheme was to study its effects on the aerosol size distribution and its potential further effects on cloud droplet size distribution.

  Any suggestions what other aqueous-phase chemical reactions would be interesting from the atmospheric chemistry standpoint are greatly appreciated.

- **P. 2, ll. 28-29: What about the collection of submicron aerosol by droplets (e.g., Ardon-Dryer et al. (2015), Ladino et al. (2011))? These processes are not considered in the current model but also affect the size of aerosols.**

  The particle-based method is very well suited for studying such process. The deliquescent aerosol particles and droplets are represented in the same way in the super-droplet method - there is no artificial category of "aerosol particle", "cloud droplet" or "rain drop". The only change needed to resolve the collection of submicron aerosol particles by droplets in the current setup is adding a collision efficiency look-up table based on the works of Ladino et al. (2011) and Ardon-Dryer et al. (2015) that would correctly represent the probability of collision between submicron aerosols and bigger drops.

  We have added this process to the list of cloud-related processes that affect the aerosol sizes. We also added information on how it should be included in the particle-based microphysics scheme in Sec. 2.

- **P. 3, l. 2: What is meant by an atmospheric aerosol particle in this context? There must certainly be a size restriction to the cited statement since very small aerosols are not cloud-active.**

  The mentioned study serves as an estimate of the global average recycling by clouds. From our understanding of the derivation of those estimates, the relevant aerosol sizes are between $0.1 \mu m$ and $1\ \mu m$.

- **P. 3, ll. 5-6: The effect of collision and coalescence on the amount of aerosol inside a cloud droplet (in absence of aqueous sulfur chemistry!) has been considered in an LCM framework before (see Hoffmann, 2017).**

  We are mentioning this work in Sec. 2.

- **P. 4, ll. 11-12: This is a vague statement since it depends on the equations considered and the applied numerical solvers.**

  We crossed out this sentence.

- **P. 5, l. 2: How does the wind speed affect the chemical scheme?**

  The scheme requires the velocity field (or wind field) to be provided, in order to advect the super-droplets in the computational domain. Because the scheme can be used in models of different dimensionalities, this

velocity field could either be a constant number (as in parcel model simulations in Sec. 4) 2-dimensional prescribed flow field (as in the 2-dimensional simulations in Sec. 5) or a 3-dimensional LES velocity field (hopefully in the future). To clarify the sentence we changed the "wind speed" to "wind field".

- **P. 5, ll. 8-13: Does this restriction affect the results? And if yes, how? And what happens to giant and ultra-giant aerosols, which remain less diluted throughout their lifecycle?**

  The assumption that the solution droplets are diluted only affects the chemical calculations of the presented scheme. The impact of GCCN on rain formation could be studied from the collisions perspective only. No changes to the current version of the scheme would be necessary for such simulations. To study the impact of GCCN on the aqueous phase chemistry another aerosol types should be added first. For example, the sea-salt aerosol particles and chemical processes relevant to them should be included, see the discussion starting in line 30 on page 13.

  In the simulations discussed in the manuscript, the assumption that solution droplets are diluted excludes from the aqueous phase simulations deliquescent aerosol particles and rain drops at the last stages of their evaporation. In both cases, excluding aqueous phase oxidation reaction is very beneficial for the condensation/evaporation scheme. Especially during activation of cloud droplets the wet radius of droplets changes very fast. It would be difficult for the condensation scheme to reach convergence during cloud droplet activation if the dry and wet radii of the super-droplets were allowed to change simultaneously.

  We tested the dependence of the parcel model simulations for the threshold values of 0.01 moles/liter and 0.1 moles/liter. In both cases we found no significant difference from the 0.02 moles/liter threshold that is used by default. Any further tests should be done in a full 3-dimensional LES setup that would allow for a longer simulation times with a more realistic representation of rain formation and aerosol recycling by clouds. However such simulations are out of the scope of the current paper.

- **P. 5, ll. 20-22: In what aspect is the LCM beneficial? Using a low number of super-droplets, the LCM will suffer from the same problems as a moving bin scheme.**

  We crossed out this statement.

- **P. 5, Eq. (1): For a reaction ab < − > a + b dissociation constant is defined as K = [a][b]/[ab]. Accordingly, Eq. (1) is just a relationship to adjust a measured dissociation constant K to a given temperature. Denote the meaning of T and R.**

  Done!

- **P. 6, Eq. (2): I understand that this equation is based on the neutral charge condition. For clarity, it might be necessary to comment on how the expressions for the individual terms are derived. (Or state an appropriate reference.)**

  We added a reference to the relevant chapter in Seinfeld and Pandis (2016).

- **P. 6, ll. 15-16; p. 7, l. 15: How is this small time-step determined? Is there a criterion? How does the user know that an appropriate time-step is chosen?**

  Unfortunately we don't have a criterion to determine the necessary time-step. The system is non-linear and the exact criterion depends on the changes in droplet radius due to condensation and collisions as well as the actual chemical composition of the droplets. The processes limiting the time-step length in the simulations presented in the manuscript are the condensation/evaporation and the aqueous phase chemistry. The parcel model simulations from Sec. 4 and the 2-dimensional simulations from Sec. 5 use model time-step equal to 1 second. For the simulations from Sec. 5 additional 10 sub-steps were used per each model time-step for both condensation and aqueous phase chemistry schemes.

  The *libcloudph++* can be compiled and run in the debugging mode. In this mode all the asserts (for example checks if the trace-gas fields are non-negative) are performed. When using the library in a new modeling setup it is worthwhile to first use it in a debugging mode to check that the simulation is stable and if necessary adjust the model time-step.

- **P. 7, ll. 26-28: Why is H2SO4 used instead of SIV to denote sulfuric acid? Please clarify.**

  We changed it to SVI.

- **P. 8, ll. 4-5: What is meant by initial conditions above supersaturation?**

To clarify, we changed it to "initial conditions where supersaturation is present in the environment". In general, such initial conditions are used in the kinematic model setups.

- **P. 8, ll. 15-20; p. 12, Fig. 3: How is the aerosol distribution transformed to super-droplets? Is the distribution binned and a super-droplet represents each bin or is a random generator used that follows the given distribution? This is in important information to understand the slight fluctuations in the initial spectrum (Fig. 3).**

The dry radius of super-droplets is chosen randomly with a uniform distribution in the logarithm of the radius. The minimal and maximal values of the dry radius are chosen automatically by evaluating the initial size distribution. The criterion is that the super-droplet multiplicity has to be greater or equal to 1. We added the reference to Sec. 5.1.6 in Arabas et al. (2015) where the procedure is discussed in detail.

- **P. 10, ll. 9-10: How is it possible to have activated particles at a relative humidity of 100 %? To activate a droplet, a supersaturation ($RH > 1$) is necessary.**

Agreed. We crossed out this statement and only report the number of cloud droplets after the maximum super-saturation is reached.

- **P. 13, ll. 3-4: The Hall (1980) collection efficiencies are not very suitable for big droplets since they neglect the effect of a coalescence efficiency of less than unity. How are big droplets defined?**

The collision efficiency values used in the simulations presented in the manuscript are shown in Fig. 2. Left panel shows the collision efficiencies based on Hall (1980). They are used for droplets with wet radius greater than 20 $\mu m$. Right panel shows the collision efficiencies based on Pinsky et al. (2008), which are used for collsions between droplets with radius smaller than 20 $\mu m$. For easier comparison, the middle panel of Fig. 2 zooms in on the Hall (1980) collision efficiencies for the small droplets.

In general, the 2-dimensional simulations presented in Sec. 5 of the manuscript do not produce very big rain drops. At the end of the simulation the concentration of rain drops with radius greater that 100 $\mu m$ does not exceed 0.01 per mg of dry air. The rain drop sizes are limited by the short simulation time and 2-dimensional kinematic setup. Different collision kernels and efficiencies should be tested for longer 3-dimensional simulations.

[Figure]

Figure 2: The collision efficiencies used in the presented simulations. The left panel shows the Hall (1980) collision efficiency. The middle panel zooms-in to show the Hall (1980) collision efficiency for small droplets. The right panel shows the collision efficiency for small droplets based on Pinsky et al. (2008).

- **P. 16, Fig. 4b: What is meant by spec. in the panels title?**

It was supposed to mean specific concentration - i.e. concentration defined per unit mass of dry air. We have updated the caption of Fig. 4b.

- **P. 18, l. 5: Eq. (5) shows that the oxidation of H2O2 is not independent of the acidity ([H+] in the denominator). (Although the dependence might be very weak.)**

Corrected.

- **P. 18, l. 19-20: This is because ... aerosol particles. Is this a direct result of Eq. (2)?**

Yes. But it also depends on the speed of the dissolution of trace gases. We added the reference to Eq. (2).

- P. 1, ll. 20-21: clearly defined user interface is a slightly subjective statement.
  P. 1, l. 21: Make clear that separation of concerns is a software design principle.
  P. 2, l. 17: Define IPCC.
  P. 2, ll. 33-34: All these references should be in one pair of parentheses.
  P. 3, l. 20-22: As a service to the reader, it would be nice to state how many new attributes are added for the consideration of aqueous sulfur chemistry.
  P. 4, l. 15: The citation is Lee et al. (2014) not Junghwa et al. (2014).
  P. 4, ll. 22-23: The pH is a scale used to quantify the acidity of a solution. It is (linguistically) awkward to keep track of a scale and not the underlying physical meaning.
  P. 6, Eq. (2): The underbrace needs to include all terms on the right-hand side of the equation.
  P. 10, l. 4: These values are stated in Tab. 2 of Kreidenweis et al. (2003).
  P. 12, ll. 7-8, 15: Semicolons in an enumeration of references are unusual.
  P. 13, l. 21: the the
  P. 15, l. 27, 29: Is Fig. 4f addressed of Fig. 4c?

All lines have been corrected. Thank you!
- **p. 7, l. 14: The sentence is not clear. Should it read This approach does ensure. . . the total dissolved mass. . .does not exceed?**

  No. The dissolution of trace gases is calculated individually for each super-droplet. The trace gas mixing ratio is not updated after each super droplet dissolves the required trace gas mass. Instead, at the end of the dissolution calculation, the sinks from all super-droplets are summed and then applied to the ambient trace gas mixing ratio. In principle, when used with long model time-steps, this could result in dissolving more than the total trace gas mass available in the environment. To prevent that, short model time-steps or even sub-steps are required. The time-step length required by the dissolution scheme is similar to the time-step required by the condensation scheme.

  The according sentence in the manuscript was corrected.

- **p. 4, l. 8: A more detailed discussion of the errors would be useful to better appreciate the usefulness of your framework. How large are the statistical errors as compared to the numerical diffusion errors in previous schemes?**

  Unfortunately, it is difficult for us to provide an estimate of the total error due to the numerical diffusion in the bin schemes referenced in this manuscript. It depends on the number of bins used in the scheme, as well as the actual numerical algorithm used to integrate the equations.

  The study by Li et al. (2017) provides a good comparison between the bin and particle-based schemes. As mentionned in the manuscript, the study by Unterstrasser et al. (2017) tests the behavior of different implementations of collisions in the super-droplet scheme. Additionally, Dziekan and Pawlowska (2017) tested the accuracy of the collision scheme in the super-droplet method when compared to the solutions of master equation, direct numerical simulations and the Smoluchowski equation. They found that for high super-droplet concentrations, the super droplet method resolves the collisions between droplets very accurately.

  Interestingly, the work by Grabowski and Abade (2017) suggests that the fluctuations of the supersaturation should be included in super-droplet method and will result in an additional broadening of the droplet size distribution. In contrast to the broadening due to numerical diffusion in the bin schemes, such process can be accurately modeled in the super-droplet method when the turbulence dissipation rate in the model grid-box is known.

- **p. 2, l. 1-6: This paragraph should be moved to the end of the introduction.**
  **p. 2, l. 2: It should be also mentioned here that uptake processes between the gas and aqueous phases are included. As it is, it reads that only aqueous phase chemistry is considered but the full multiphase system is implemented.**
  **p. 2, l. 10: SO2 can be oxidized within minutes or a few hours within clouds.**

p. 5, l. 29, and throughout manuscript: deltaH/R are not correction coefficients. They should be called temperature dependence or enthalpy of ionization (= only deltaH).

p. 7, l. 5, and throughout manuscript: The equilibrium dissolution constants are commonly referred to as Henrys law constants. Their temperature dependencies are enthalpy of solution.

p. 7, l. 25: setup misspelled

p. 7, l. 29: Add unit M to [H+]

p. 7, l. 30: Why do you refer here to Table C2?

p. 9, l. 2, and throughout manuscript: water weighted average might be misleading. I assume you mean (water) volume weighted average?

p. 9, l. 8, and throughout manuscript: I suggest using S(VI) instead of H2SO4.

p. 15, l. 31: new aerosol particles is misleading. It implies new particle formation. Please clarify.

p. 17, l. 7: minimize

p. 19, l. 7/8, and 14/15: These sentences sound awkward as SVI molecules are not oxidized. Try 92% of S(VI) originates from S(IV) oxidation by H2O2 (or similar).

p. 19, l. 20: Not clear what relative importance refers to here.

All lines have been corrected. Thank you!

**Additional changes to the manuscript:**

- We changed the library version number from 1.1 to 2.0. The change reflects the new library programming interface related to the aqueous phase chemistry. The final version of the manuscript will be accompanied by the corresponding code release on GitHub

- To simplify we have removed from the abstract " (...) which allow reusing the created scheme from models implemented in other programming languages".

- p. 2, l.33: We added a list of names used to describe this process.

- p.14: We have deleted the last sentence.

- We have changed the Appendix A section name from User Interace to Programming Interface

- We have updated the list of references.

**References**

[revised manuscript text omitted]

---

## Author Response (AR2)

We would like to thank the Editor for his comments on the manuscript. All comments have been addressed in the revised version of the manuscript. Throughout the manuscript we have tried to fix our articles usage to the best of our ability. Below we are submitting the revised version of the manuscript.

[revised manuscript text omitted]